# Overview of Time Synchronization for IoT Deployments: Clock Discipline Algorithms and Protocols

**DOI:** 10.3390/s20205928

**Published:** 2020-10-20

**Authors:** Hüseyin Yiğitler, Behnam Badihi, Riku Jäntti

**Affiliations:** Department of Communications and Networking, Aalto University, 02150 Espoo, Finland; behnam.badihi@aalto.fi (B.B.); riku.jantti@aalto.fi (R.J.)

**Keywords:** internet of things, capillary networks, wireless sensor networks, clock synchronization, clock discipline algorithms, time synchronization protocols

## Abstract

Internet of Things (IoT) is expected to change the everyday life of its users by enabling data exchanges among pervasive things through the Internet. Such a broad aim, however, puts prohibitive constraints on applications demanding time-synchronized operation for the chronological ordering of information or synchronous execution of some tasks, since in general the networks are formed by entities of widely varying resources. On one hand, the existing contemporary solutions for time synchronization, such as Network Time Protocol, do not easily tailor to resource-constrained devices, and on the other, the available solutions for constrained systems do not extend well to heterogeneous deployments. In this article, the time synchronization problems for IoT deployments for applications requiring a coherent notion of time are studied. Detailed derivations of the clock model and various clock relation models are provided. The clock synchronization methods are also presented for different models, and their expected performance are derived and illustrated. A survey of time synchronization protocols is provided to aid the IoT practitioners to select appropriate components for a deployment. The clock discipline algorithms are presented in a tutorial format, while the time synchronization methods are summarized as a survey. Therefore, this paper is a holistic overview of the available time synchronization methods for IoT deployments.

## 1. Introduction

Recent advances in embedded intelligence, connectivity, and interaction technologies have allowed integrating pervasive objects from our daily life into communication networks to interact with each other over the Internet for enabling novel applications and services. This emerging communication and computing paradigm is often referred to as Internet of Things (IoT), and it utilizes the Internet as both communication and virtualization platform to link the physical world to the information (virtual) world [1]. The broad interconnection possibilities supported by the IoT brings forth interoperability problems between different objects with heterogeneous capabilities [2]. In a typical IoT deployment, three different networks are taking part as shown in Figure 1. A node of a wide area network (WAN), e.g., user entity of a cellular network, is connected to a node of a local area network (LAN) over a network interface controller (NIC), and this node is connected to a personnel area network (PAN) of a low-power and short-range wireless communication technology (Such an extension of a WAN network toward local and personal area networks by using a WAN node as a backhaul connection entity is also known as a capillary network [3]). Some of these objects may possess a large amount of computational and communication resources, some may be energy constraint wireless sensor nodes, and some may be passive simple devices such as RFID tags. Although these components can be, in principle, interconnected using contemporary Internet technologies, not all objects can accommodate the resources required by these solutions. One approach is to use translation entities between these devices and an interrogator to connect to the Internet, e.g., a gateway [4]. However, this approach introduces processing and translation overhead, which alters the yield of applications requiring tight interaction with the physical world. In this article, such applications are considered and wireless sensor network entities are evaluated as objects interacting with the physical world.

*Wireless sensor networks* (WSNs) [5] are one of the enabling technologies of IoT. They transmit acquired digital data from the physical world to the Internet or conversely receive data from the Internet that describe actions to make some changes to the surroundings in order to reach a common goal without human intervention [6]. This is achieved by seamlessly conveying the information content by performing various translations between the involved communication and processing entities. In this regard, applications requiring chronological information ordering or synchronous execution for data fusion, or low-power networking and time-division transmission scheduling [7,8], require a coherent notion of *time*, which must be shared among all the objects taking part in both processing and communication.

The notion of the time of a clock is different than the universal time due to several factors as discussed by Allan [9]. The impact of these errors is mitigated by:
(i)transmitting a time report of a reference clock using a messaging protocol;(ii)mitigating non-determinism in message delivery and time measurements using a Clock Discipline Algorithm (CDA), and;(iii)adjusting the clocks.

For a typical IoT deployment, there are three different networks that have different notions of time as shown in Figure 1. For WANs and LANs, the time synchronization problem has well-known solutions such as the Network Time Protocol (NTP) [10,11] (or its tailored version the Simple Network Time Protocol [12]), and Precision Time Protocol (PTP) [13]. For example, the WAN node (and also the LAN nodes) may synchronize to the Internet time, e.g., using the NTP. However, the computation and energy requirements of these solutions cannot be fulfilled by objects with constrained computational and energy resources, and such objects call for simpler methods. The existing time synchronization solutions for WSNs are designed for constrained devices (e.g., the PAN network in Figure 1), but their performance is tailored for a specific application scenario. For the deployment in Figure 1, the global notion of time in WAN and LAN nodes must still be disseminated to the PAN nodes through the radio attached to CPU3, and there are no readily available solutions to fulfill accuracy requirement of all application scenarios. Therefore, IoT practitioners are required to select a suitable time synchronization solution considering the application scenario and the capabilities of the network entities to achieve the required level of performance.

In this work, we present a comprehensive survey of time synchronization methods for IoT deployments in the following organization. In Section 2, some motivating example applications requiring a coherent notion of time are given, and available survey works are presented. Thereafter, the individual components of time synchronization are discussed starting from clock models in Section 3. The solutions of the clock synchronization problem, known as CDA, are presented in Section 4. In   Section 5, several aspects of time synchronization messaging are elaborated on, and two example empirical time data are used in Section 6 to demonstrate the impact of time error sources. Finally, conclusions are drawn in Section 7.

## 2. Background

In this section, we first introduce a general IoT platform, and then summarize several applications that require a coherent notion of time. Thereafter, we review available time synchronization survey  works.

### 2.1. IoT Platform

IoT is on the verge of changing the traditional concept of connectivity for everyone to connectivity for everything anytime and anywhere [14]. It is a radically large and highly dynamic distributed system with a massive number of entities producing and consuming information [15] to form a *common operational picture* (COP) for different (novel) applications and services [6] as shown in Figure 2. The objects in IoT are by no means limited to the entities that are directly connected to the Internet. On the contrary, any uniquely identifiable physical entity, which may be connected to the Internet over a gateway (interrogator) [16], is allowed. Therefore, IoT realizes ubiquitous computing and networking by making the benefits of the technology inseparable part of the daily living environment [17], which  is expected to unprecedentedly alter the behavior of its users [18].

The information content of the data generated by an IoT network is used by applications and services through the COP. Semantically, the COP does not discriminate between real objects and virtual (information) objects, and it is natural to virtualize all the information sources and sinks as shown in Figure 3. These entities are called *virtual objects*, and they may proxy physical things or they may be linked to a software component. In either case, they represent a unit of information in the COP. Such an information-based abstraction, in return, requires a consensus in ordering the data with respect to a specific argument, e.g., time or frequency. For most of the physical information sources, the acquired data is naturally ordered in time so that the information content assigned to different virtual objects must be ordered with respect to their chronology. In particular, this is important in the following aspects [19]:(i)Querying the universal time at which a specific event happened and observed by an object.(ii)Measuring the time difference between two events that are observed by different objects.(iii)Relatively ordering the events that are observed by different objects.

Therefore, all the physical information sources must have a common notion of time to fuse all the virtual objects into the COP.

### 2.2. Motivation

In order to build virtual objects using implicit measurements of some phenomenon, the output of different sensors is usually fused. The acquired data from different sources are combined and correlated to obtain additional insight, which is usually not evident in the original data [20]. The cross-correlation of data from different sources must be calculated using the acquired samples, which requires a shared notion of time. Therefore, implicit information sources might rely on a common notion of time among the physical objects.

The distributed nature of WSNs allows acquiring samples from a spatio-temporal field, e.g., for structural health monitoring [21]. These applications require collecting synchronous samples from all the sensors to estimate the spatial parameters of interest [22]. Therefore, the synchronous execution of certain periodic tasks, e.g., sampling, requires a coherent notion of time within the network.

Some local deployments of WSNs require a guarantee to complete a task within a certain time limit, e.g., for industrial automation [23], recently referred to as industrial wireless sensor networks [24]. Low-jitter applications require estimation and compensation of various time uncertainty sources, including the ones originating from oscillators and the communication entities. For these use cases, the time-synchronized operation must be provided by the underlying networking technology, as it is for WirelessHART or ISA100.11a standards [25]. Both of these protocols form their network using the Time Synchronized Mesh Protocol [26], which can only be realized with tightly synchronized nodes. These systems have recently led to the development of Time Synchronized Channel Hopping (TSCH) networks [27], which are built upon the IEEE 802.15.4e standard [28], and have IPv6 support based on several IETF standards [29]. For these networks, the time synchronization also enables various techniques to improve reliability in terms of packet delivery ratio. Furthermore, the nodes are allowed to turn-off their power-hungry components when they are not needed, which significantly decreases power consumption. Thus, time-critical, reliable, and low-power networking solutions require tight time-synchronization.

The advances in Mobile Adhoc Networks (MANET)s have enabled Vehicular Adhoc Networks (VANET)s [30], which aim at realizing Intelligent Transportation Systems by supporting both safety-related applications to reduce the probability of traffic accidents, and non-safety applications to improve passenger comfort and accommodate commercial platforms. These networks are characterized by their intermittent connectivity and high network node speed, and the success of just mentioned applications depends on the accuracy of the notion of time of each vehicle [31]. In general, VANETs can acquire precise global time from satellite-based navigation systems [32], but this may fail when the vehicles are out of the satellite system coverage. In these cases, a global notion of time can be disseminated in the network using the time synchronization methods for the IEEE 802.11 [33] since the network operation is mainly based on the IEEE 802.11p [34]. Therefore, a successful realization of intelligent transportation systems depends on the accurate notion of time within VANETs.

The management and distribution of an electricity grid using communication and pervasive computing technologies have recently gained momentum to support distributed renewable energy sources, which is often referred to as Smart Grid [35]. A wide-area monitoring system, fault detection, protection functions, substation monitoring, and fault recording all require a network-wide shared notion of time [36]. Therefore, Smart Grid IoT deployments depend on an accurate and coherent notion of time within their network.

With the advent of the IoT in the industrial domain, which is also known as industrial IoT (IIoT), the deployments supporting time-aware and precise timestamped operations have gained much interest. These distributed systems are implemented using low-cost devices with the ability to sense and monitor the physical phenomenon, and they have been deployed in the food chain industry, industrial automation and agriculture. Most notably, these applications require low-latency communication, precise data processing, and trusted services [37], which can only be achieved by network-wide, accurate, and robust time-synchronization. In the remaining part of the article, we  elaborate on several aspects of time synchronization that can be used also in IIoT applications.

### 2.3. Related Works

The time synchronization in WSNs is one of the widely studied problems, and several survey works exist. A summary of the available surveys is given in Table 1. The surveys [19,38,39,40] aim at aiding practitioners to select or develop time synchronization protocols, and their scope is mainly restricted to messaging schemes. In particular, a comprehensive survey by Sundararaman et al. [19] aims at showing the link between time synchronization for WSNs, and distributed systems and wired networks. Similarly, the works [41,42] provide classification methodologies for the available protocols. The time synchronization problem in the signal processing perspective is studied in the survey by Wu et al. [7], where the main scope is on exponentially distributed delivery delays. Network-wide synchronicity and coupled-clock’s based synchronization solutions are studied in the surveys by Simeone et al. [43], and Bojic and Nymoen [44]. The book by Serpedin and Chaudhari [45] is a complete reference for estimation methods, and time synchronization protocols presented until 2009. It can be concluded from the summary provided in the table that the scope of the available survey works are constraint to certain topics of the general time synchronization problem. However, IoT networks require one to take into account the transition of the notion of time among different networks, which require designing different methodologies for different applications. Therefore, there is a need for an overview survey, which is not restricted to a particular problem but presents each component and associated solution in a bottom-up approach. In this effort, we outline individual components taking part in the synchronization, starting from oscillators, and discuss the advantages and limitations of available methods in the IoT deployment perspective.

In this work, we provide both a tutorial-like summary of the clock models and CDAs, and  a comprehensive survey of the time synchronization protocols. Different than the other available surveys, the provided clock relation models are derived step by step, and various CDAs are derived for different models. The derived clock relation model is more general than the available models, and it shows that the time reports of software clocks are correlated in time. This correlation not only degrades the performance of well-known estimators but also bounds the synchronization period. The derived model is used for developing a computationally light-weight recursive clock discipline algorithm, which is consistent and efficient. The time synchronization protocol survey aims at providing an overview of different time synchronization components including timestamping, messaging schemes, multi-hop  approaches, and several networking practical issues and their available solutions. A discussion on the presented components is given to clarify the advantages of the methods. Therefore, this work aims at aiding practitioners to select appropriate clock synchronization components in the complete IoT deployment illustrated in Figure 1.

The time synchronization is one of the widely studied problems in computer networks. In  this work, we focus primarily on the time synchronization for low-cost and low-power networks, and give a summary of synchronization methods for LAN and WAN networks. Detailed surveys on different time synchronous networks are tabulated in Table 2, and the reader is referred to these articles. However, this work may still provide valuable insights on the clock models, clock discipline algorithms, and underlying protocol primitives that affect the synchronization accuracy.

## 3. Clock Models

In this section, we elaborate on the clock relation models. We first introduce the parameters defining a software clock and then derive a model that relates the reports of one clock in terms of the reports of another. Finally, we summarize the widely used clock relation models.

### 3.1. Software Clocks

A *clock* measures the time elapsed since an epoch. Although ideal clocks report $C_{\ell }\left(t\right)=t$ at the universal time instant *t*, practical clocks can only report their time at discrete instances since they are usually implemented as a counter driven by an oscillator [45] as shown in Figure 4. The most important impact of such an implementation on the time reports of the clock is their deviation from the actual time due to the imperfections of the driving oscillator. This variation is modeled as a second-order polynomial with coefficients *frequency drift*
ω, *frequency offset*
γ, *time-offset*
θ, and random variations ε [9]. Consequently, the *time deviation* of a continuous clock, say $C_{\ell }\left(t\right)$, at instant *t*, is given by
(1)$$C_{\ell }\left(t\right)-t=\theta_{\ell }+\gamma_{\ell }t+\omega_{\ell }t^{2}+\varepsilon_{\ell }\left(t\right),$$
where subscripts identify the clock under consideration.

The time deviation due to oscillator imperfections can be observed in the frequency spectrum of a practical oscillator’s output, which is spread around localized tones rather than being non-zero only at discrete frequencies. This phenomenon can be modeled using an ordinary differential equation with a periodic solution, which has a *phase deviation* (a.k.a. time jitter) due to random perturbations. For uncorrelated perturbation sources, the phase deviation is a Wiener process as previously shown by Demir [49]. This result is used in previous work [50] to show that $\varepsilon_{\ell }\left(t\right)$ can be taken as a zero-mean Gaussian with variance $\sigma_{\ell }^{2}\left(t\right)$ and with auto-correlation function $R_{\ell }\left(t_{1},t_{2}\right)$ given as
(2a)$$\begin{array}{cc} \; & \sigma_{\ell }^{2}\left(t\right)={\left(1+\gamma_{\ell }\right)}^{2}c_{\ell }t, \end{array}$$
(2b)$$\begin{array}{cc} \; & R_{\ell }\left(t_{1},t_{2}\right)=c_{\ell }\min\left\{t_{1},t_{2}\right\}, \end{array}$$
where $c_{\ell }$ is the oscillator variance constant.

The model in Equation (Equation 1) implies also that the time error at the *k*^th^ transition of the edge-triggered counter is
(3)$$e_{\ell }\left[k\right]=t-kT_{\ell }=t-\frac{{\bar{T}}_{\ell }}{1+\gamma_{\ell }}k,$$
where ${\bar{T}}_{\ell }$ is the nominal period of the oscillator, and $T_{\ell }≜{\bar{T}}_{\ell }/\left(1+\gamma_{\ell }\right)$ is the instantaneous period. Due to the nature of the underlying uncertainty sources discussed above, $\varepsilon_{\ell }\left(t\right)$ is a zero mean Gaussian process [50]. Respectively, the time error $e_{\ell }\left[k\right]$ is also a zero-mean Gaussian process with approximate second-order statistics given as
(4a)$$\begin{array}{cc} \; & E\left\{e_{\ell }^{2}\left[k\right]\right\}\approx {\left(1+\gamma_{\ell }\right)}^{2}c_{\ell }t_{k}, \end{array}$$
(4b)$$\begin{array}{cc} \; & E\left\{e_{\ell }\left[k_{1}\right]e_{\ell }\left[k_{2}\right]\right\}\approx c_{\ell }\min\left\{t_{k_{1}},t_{k_{2}}\right\}, \end{array}$$
where $t_{k}$ denotes the universal time when the *k*^th^ sample is acquired, and E{·} denotes the statistical expectation.

### 3.2. Clock Relation Model

The time deviation model in Equation (Equation 1) can be used for relating the reports of two clocks, say the reference clock $C_{1}\left(t\right)$ and the other clock $C_{2}\left(t\right)$, since the universal time *t* is common. If there is a message delivery delay d(t) measured in the universal time associated with the transfer of the reference clock report $C_{1}\left(t\right)$ as visualized in Figure 5, then the time report of $C_{2}\left(t\right)$ when the report of $C_{1}\left(t\right)$ is received is given by
(5)$$\begin{array}{cc} \; & C_{2}\left(t\right)=\alpha_{12}C_{1}\left(t\right)+\theta_{12}+\varepsilon_{12}\left(t\right)+\left(1+\gamma_{2}\right)d_{12}\left(t\right),\\ \; & \alpha_{12}≜\frac{1+\gamma_{2}}{1+\gamma_{1}},\;\;\theta_{12}≜\theta_{2}-\alpha_{12}\theta_{1},\;\;\varepsilon_{12}≜\varepsilon_{2}-\alpha_{12}\varepsilon_{1}, \end{array}$$
where $\alpha_{12}$ is the *clock skew* between $C_{1}\left(t\right)$ and $C_{2}\left(t\right)$, and the frequency drift terms $\omega_{1}$ and $\omega_{2}$ are ignored, following the common practice in the literature. Equation (Equation 5) is known as the *clock relation model* [45], and some of its parameters are visualized in Figure 5. In the following, the clock relation models are gradually developed.

#### 3.2.1. Clocks on the Same Processor

Suppose that we are aiming to compare the time reports of clocks $C_{1}\left(t\right)$ and $C_{2}\left(t\right)$ that are implemented on the same processor, that is $d_{12}\left(t\right)=0$, and both are initialized to zero a t=0. The relation between the count values of the counters associated with these clocks can be written as
(6)$$\begin{array}{cc} \; & m=\alpha_{12}\frac{{\bar{T}}_{1}}{{\bar{T}}_{2}}k+e\left[m\right],\\ \; & e\left[m\right]≜\frac{1}{T_{2}}\left(e_{1}\left[k\right]-e_{2}\left[m\right]\right), \end{array}$$
where *m* is the count value of the counter of the clock $C_{2}\left(t\right)$ and *k* is the count value of the counter of the clock $C_{1}\left(t\right)$. The error term e[k] is also Gaussian since $e_{1}\left[k\right]$ and $e_{2}\left[m\right]$ are independent and Gaussian. The second order statistics of e[k] can be derived using the statistical independence of $e_{1}\left[k\right]$ and $e_{2}\left[m\right]$, which yields
(7a)$$\begin{array}{cc} \; & E\left\{e^{2}\left[m\right]\right\}\approx \frac{1}{{\bar{T}}_{2}^{2}}\left(\alpha_{12}^{2}c_{1}+c_{2}\right)t_{m}, \end{array}$$
(7b)$$\begin{array}{cc} \; & E\left\{e\left[m_{1}\right]e\left[m_{2}\right]\right\}\approx \frac{1}{T_{2}^{2}}\left(\alpha_{12}^{2}c_{1}+c_{2}\right)\min\left\{t_{m_{1}},t_{m_{2}}\right\}. \end{array}$$

The error term e[m] is zero mean if and only if the clock periods are an integer multiple of one another, and they are phase locked so that at a universal time instant *t* both of the edge counters have just incremented. This also implies that the zero mean Gaussian assumption is valid if and only if the residual time error associated with the counter increments is negligible.

For the discrete time relation model given in Equation (Equation 6), the nominal clock periods ${\bar{T}}_{1}$ and ${\bar{T}}_{2}$ are known constants of the oscillators, and the clock offset is zero since the clocks are implemented on the same processor. Hence, the synchronization problem is to estimate the clock skew $\alpha_{12}$. Let us suppose that *N* time reports of both of the clocks are acquired to estimate $\alpha_{12}$. For notational convenience, let us define the following vectors and matrices,
$$\begin{array}{cc} \bar{k} & ={\left[\begin{array}{cccc} k_{1} & k_{2}-k_{1} & \cdots & k_{N}-k_{N-1} \end{array}\right]}^{⊤},\\ \bar{m} & ={\left[\begin{array}{cccc} m_{1} & m_{2}-m_{1} & \cdots & m_{N}-m_{N-1} \end{array}\right]}^{⊤},\\ \bar{e} & ={\left[\begin{array}{cccc} e[1] & e[2]-e[1] & \cdots & e[N]-e[N-1] \end{array}\right]}^{⊤},\\ k & =U\bar{k},\;m=U\bar{m},\;e=U\bar{e}, \end{array}$$
where ${}^{⊤}$ denotes the matrix transpose and U is a lower triangular matrix defined as
$$U=\left[\begin{array}{cccc} 1 & 0 & \cdots & 0\\ 1 & 1 & \cdots & 0\\ \vdots & \vdots & \ddots & \vdots \\ 1 & 1 & \cdots & 1 \end{array}\right].$$

*Progressive clock relation model:* Using the definitions given above, the instantaneous count values are related to each other with
(8)$$m=\alpha_{12}\frac{{\bar{T}}_{1}}{{\bar{T}}_{2}}k+e.$$
Since the time reports of clocks are monotonically increasing, this model is referred to as a *progressive time relation model*, and it naturally follows from the definition of the involved quantities. For this model, the covariance matrix of the noise term e is composed of components $Q=[q_{ij}]$ given as
(9)$$\left[q_{ij}\right]=\frac{1}{T_{2}^{2}}\left(\alpha_{12}^{2}c_{1}+c_{2}\right)\min\left\{t_{i},t_{j}\right\}.$$
Therefore, the progressive time relation model in Equation (Equation 8) has a correlated noise term.

*Incremental clock relation model:* Since the lower triangular matrix U is invertible for all non-zero *N*, it follows from the definition of m, k and e that
(10)$$\bar{m}=\alpha_{12}\frac{{\bar{T}}_{1}}{{\bar{T}}_{2}}\bar{k}+\bar{e}.$$
This time relation model operates on the incremental time reports of the clocks, and it is referred to as the *incremental clock relation model*. For this model, the covariance matrix of the error term $\bar{e}$ is a diagonal matrix with components $\bar{Q}=\left[{\bar{q}}_{ij}\right]$ given as
(11)$$\left[{\bar{q}}_{ij}\right]=\left\{\begin{array}{cc} \frac{1}{T_{2}^{2}}\left(\alpha_{12}^{2}c_{1}+c_{2}\right)\left(t_{i}-t_{i-1}\right) & i=j,\\ 0 & i\neq j. \end{array}\right.$$
This diagonal matrix indicates that the incremental model in Equation (Equation 10) has independent noise samples, enabling optimal performance for well-known estimators.

*Time offset:* In case the counters are reset to zero at t=0, the clocks are related to each other through a single parameter $\alpha_{12}$. In order to relax this assumption, let us now suppose that the initial count values of the counters are $k_{0}$ and $m_{0}$. Then, the progressive model in Equation (Equation 8) can be written as
(12)$$m=\alpha_{12}\frac{{\bar{T}}_{1}}{{\bar{T}}_{2}}k+\left(m_{0}-\alpha_{12}\frac{{\bar{T}}_{1}}{{\bar{T}}_{2}}k_{0}\right)1+e,$$
where 1 is all one vector. On the other hand, for the incremental model in Equation (Equation 10), the new time-offset term only appears in the first time report pair $\left(m_{1},k_{1}\right)$, that is once the time-offset is compensated for, the only parameter that needs to be estimated and compensated for is $\alpha_{12}$. Therefore, the time incremental model eliminates the need for jointly estimating the time-offset and the clock skew, at the cost of implementing two estimators for each parameter.

#### 3.2.2. Clocks on Different Processors

If the clock being synchronized is implemented on a different processor, there is a non-zero time report delivery delay $d_{12}\left(t\right)$. This delay depends on several message delivery implementation- dependent factors, but all can be decomposed as
(13)$$d_{12}\left(t\right)=D_{12}+\delta_{12}\left(t\right),$$
where $D_{12}$ is deterministic and constant delays, and $\delta_{12}\left(t\right)$ is the random message delay at *t*. Thus, a general clock relation model can be reached by including the messaging delay characteristics into Equation (Equation 5) as
(14)$$C_{2}\left(t\right)=\alpha_{12}C_{1}\left(t\right)+\underset{\mathrm{time-offset}}{\underset{︸}{\theta_{12}+D_{12}}}+\underset{\mathrm{random}\;\mathrm{variation}}{\underset{︸}{\varepsilon_{12}\left(t\right)+\delta_{12}\left(t\right)}}.$$

The time relation model in Equation (Equation 14) implies that the time-offset $\theta_{12}$ and the deterministic delay $D_{12}$ are not distinguishable and both contribute to the time-offset. Similarly, the observed random variations in the time reports of a clock compared to the reports of another follows the joint probability distribution of $\varepsilon_{12}\left(t\right)$ and $\delta_{12}\left(t\right)$. In other words, a general clock relation model is given by
(15a)$$\begin{array}{cc} \; & C_{2}\left(t\right)=\alpha_{12}C_{1}\left(t\right)+\tau_{12}+\epsilon_{12}\left(t\right), \end{array}$$
(15b)$$\begin{array}{cc} \; & \tau_{12}≜\theta_{12}+D_{12},\;\epsilon_{12}\left(t\right)≜\varepsilon_{12}\left(t\right)+\delta_{12}\left(t\right), \end{array}$$
where the time-offset $\tau_{12}$ and the random delay term $\epsilon_{12}\left(t\right)$ have the same impact as $\theta_{12}$ and $\varepsilon_{12}\left(t\right)$ in Equation (Equation 5). Therefore, the software clock relation models in Equation (Equation 6) have the same structure even when the message delivery delay is included in the formulation.

#### 3.2.3. Numerical Example

The time difference between clocks $C_{1}\left(t\right)$ and $C_{2}\left(t\right)$ grows with the reference time in accordance with the clock relation model in Equation (Equation 14). In order to demonstrate the significance of the involved quantities, two time series are created using the parameters in Table 3. The variation of the time reports of a local clock $C_{2}\left(t_{k}\right)$ with respect to the reports of a reference clock $C_{1}\left(t_{k}\right)$ is shown in Figure 6. For the visualized data, the reference clock reports are assumed to be transmitted every second through the message delivery scheme with Gaussian delay of parameters given also in the table. The depicted result shows that the variation of the difference between the time reports $C_{2}\left(t_{k}\right)-C_{1}\left(t_{k}\right)$ with $C_{1}\left(t_{k}\right)$ grows linearly as Equation (Equation 8) implies. Furthermore, the variance of the time difference grows as time progresses. On the other hand, the variation of the time report increments of the node $C_{2}\left(t_{k}\right)-C_{2}\left(t_{k-1}\right)$ stays small and the variance does not increase as the incremental model in Equation (Equation 10) implies. Therefore, these two models have different properties, and the clock relation model estimators for each of the models have different characteristics as it is elaborated on in the next section.

### 3.3. Available Clock Relation Models

Let us denote the clock relation model as
(16)y=R(x),
where we have defined the function argument *x* as the reference clock reading $C_{1}\left(t\right)$, and *y* as the local clock reading $C_{2}\left(t\right)$. The general clock relation model in Equation (Equation 6) is composed of several parameters: oscillator-induced noise $\varepsilon_{12}\left(t\right)$ and message-delivery uncertainty $\delta_{12}\left(t\right)$, the initial time-offset $\theta_{12}$ and deterministic message-delivery delay $D_{12}$, and the clock skew parameter $\alpha_{12}$. The relative importance of these parameters changes with the time record acquisition procedures, and different approximations are possible depending on the required level of accuracy. In the following, we give a summary of the clock relation models used in the time-synchronization literature.

#### 3.3.1. Offset-Only Model

The time relation model for this case is in the form
(17)$$R_{1}\left(x\right)=x+\tau_{12},$$
where $\tau_{12}$ is the time-offset, and the clock skew is assumed to be unity and constant, $\alpha_{12}\equiv 1$. The offset only model $R_{1}\left(x\right)$ is an under-fitted time relation model, which has a large bias although the residual variance is small Section 3.2 in [51].

Although, this model is not used for time-synchronization purposes under Gaussian random variations, it is used for exponential distributed random variations by Jeske [52], by Lee et al. [53], and by Rhee et al. [54].

#### 3.3.2. Progressive Linear Model Only with Delivery Delay

In the pioneering work by Elson et al. [55], the impact of the oscillator-induced random time deviation is ignored, and the time relation model is simplified to
(18)$$R_{2}\left(x\right)=\alpha_{12}x+\tau_{12}+\delta_{12}\left(x\right).$$

The simplified model in Equation (Equation 18) is by far the most widely used model in the literature (see e.g.,  [7,45] for a comprehensive overview). In particular, this model is used by Maróti et al. [56], and other works (e.g.,  [57,58,59]) by assuming that the random delay $\delta_{12}\left(x\right)$ is Gaussian and its samples in different messages are uncorrelated. The works [7,60] use the same model with an exponentially distributed message delivery delay.

#### 3.3.3. Incremental Linear Model Only with Delivery Delays

The progressive model in Equation (Equation 18) can be used in incremental form, which reads as
(19)$$R_{3}\left(x\right)=\alpha_{12}\left(x_{k+1}-x_{k}\right)+\delta_{12}\left(t_{k+1}\right)-\delta_{12}\left(t_{k}\right),$$
where $\delta_{12}\left(t_{k+1}\right)$ is the delivery delay term associated with $x_{k+1}$, and $\delta_{12}\left(t_{k}\right)$ is associated with $x_{k}$, when an explicit estimate of the clock skew $\alpha_{12}$ is desired. This model is first used by Hamilton et al. [61], and later by Yang et al. [62,63] in order to include the dynamics of the clock skew into the synchronization problem formulation. Since this model does not contain the time offset term, it requires a two-step clock discipline algorithm.

#### 3.3.4. High Order Models Only with Delivery Delay

The time relation model in Equation (Equation 19) can be extended by including the temporal variation of the clock skew in order to account for the ignored frequency drift parameter $\omega_{0}$ in Equation (Equation 1). This term represents slow time variations due to the supply voltage changes, the temperature fluctuations, and the aging of the oscillator [9]. One approach to take into account the frequency drift is to consider a dynamical model for the clock skew $\alpha_{12}$,
(20)$$\frac{d}{dx}\alpha_{12}=\omega_{12}+\delta_{\alpha_{12}}\left(x\right),$$
where $\delta_{\alpha_{12}}\left(x\right)$ is the random variation of the clock skew, and the clock relation model is as in Equation (Equation 19). This model is first proposed by Hamilton et al. [61], and it is later used by Yang et al. [62,63].

Models with an order higher than two have also been investigated by researchers. In the work by Kim et al. [64], the authors have studied higher order autoregressive models for clock skew, where they have also validated the model order with well-known model selection methods. The same line of reasoning has motived Masood et al. [65] to study alternative models with both open-loop and feedback terms. Such high order extensions cannot be easily linked to the well known physical clock parameters. In this regard, models exceeding the second order cannot be easily described using the terminology presented above.

#### 3.3.5. Incremental Linear Model with Delivery Delay and Oscillator-Induced
Correlation

The simplified model in Equation (Equation 18) does not take into account the correlated oscillator-induced noise. This problem does not exist for the incremental linear model, which can be written as
(21)$$R_{4}\left(x_{k},x_{k+1}\right)=\alpha_{12}\left(x_{k+1}-x_{k}\right)+\epsilon_{12}\left(t_{k+1}\right)-\epsilon_{12}\left(t_{k}\right),$$
where $\epsilon_{12}\left(t_{k+1}\right)$ is the noise term associated with $x_{k+1}$, and $\epsilon_{12}\left(t_{k}\right)$ is associated with $x_{k}$. The difference of these two terms is uncorrelated between different increments, but the variance of the measurements increases with the elapsed time between the reports.

The incremental linear model in Equation (Equation 21) can be generalized to cover a-periodic time report message delivery, which is a common problem when the time reports are exchanged over a lossy medium. For this purpose, one approach is to define the time relation model in Equation (Equation 21) as
(22)$$1=\alpha_{12}\frac{x_{k+1}-x_{k}}{y_{k+1}-y_{k}}+\frac{\epsilon_{12}\left(t_{k+1}\right)-\epsilon_{12}\left(t_{k}\right)}{y_{k+1}-y_{k}},$$
where $R_{4}\left(x_{k},x_{k+1}\right)=y_{k+1}-y_{k}$, and $x_{k}$ and $y_{k}$ are as defined above. In this case, the covariance function of $\epsilon_{12}\left(t_{k+1}\right)-\epsilon_{12}\left(t_{k}\right)$ is as given in Equation (Equation 11).

The correlations in the time increments are not generally considered in the literature with the exception of the work [50]. Such a simplification limits the performance of well-known model estimators. As we demonstrate in the next section, taking into account the time report correlations is important, and allows development of an efficient and consistent clock skew estimation algorithm. However, this model does not depend on the clock offset, and requires a two-step clock discipline algorithm development.

#### 3.3.6. Summary

Available clock relation models presented in this section are summarized in Table 4. Every model has some advantages and disadvantages, and each are useful under certain application requirements. When the application does not require tight synchronization, the offset-only model can be used. However, for other cases, a progressive first order model is usually preferred. If the application permits a higher amount of computational resources, but limited amount of communications, higher order models can be used. Although not widely used, the incremental model with delivery delay and oscillator-induced correlations can be used as a direct replacement of progressive time relation model. As we elaborate on in the next section, such a replacement enables development of an efficient and consistent model estimators.

## 4. Clock Discipline Algorithms

In this section, we consider the scenario where the time reports of a clock, say $C_{2}\left(t\right)$, are required to synchronize to the time reports of a reference clock, say $C_{1}\left(t\right)$, where *t* denotes the global time, as depicted in Figure 5. The time reports of the reference clock are conveyed to one another (The message dissemination may also be one way as discussed later.) using a connectionless protocol known as the *time synchronization protocol*. The model parameter estimation and using the estimated model to calculate the reference clock time from a local time report is referred to as *clock-discipline algorithms* (CDAs). The CDAs are developed based on the time relation models, and as the accuracy of the underlying model increases, the performance of the associated CDA increases. In the following, the CDA are developed for continuous time clock relation models. It is possible to convert the discretized time reports of the clocks to continuous time readings by assuming that the associated counter increments once in one period, and by ignoring residual time error within one period.

### 4.1. Background

A CDA is to estimate the parameters of the clock relation model in order to adjust time reports of a clock or to transform its time report to the time scale of a reference clock [7]. In this regard, the algorithm needs to estimate the clock relation parameters, i.e., the model, and then use it to predict the time reports of $C_{1}\left(t\right)$ for the given reading of $C_{2}\left(t\right)$. Let us consider the abstract clock relation model in Equation (Equation 16). For this model, the CDA first must find a best estimate $\hat{R}\left(x\right)$ of the model R(x) in some sense. Then, it must use the inverse of the estimated model $\hat{R}\left(x\right)$ to calculate its argument, that is
(23)$$\hat{x}={\hat{R}}^{-1}\left(y\right),$$
as listed in Algorithm 1 for a time relation model with both clock skew and time-offset parameters.
**Algorithm 1** Calculate synchronized time1:**Input:**  local clock report
$y=C_{2}\left(t\right)$**Global Variables:**  ${\hat{\alpha }}_{12}$
and
${\hat{\tau }}_{12}$**Output:**  Corrected time report
$\hat{x}={\hat{C}}_{1}\left(t\right)$2:**return**${\hat{C}}_{1}\left(t\right)\leftarrow \left(C_{2}\left(t\right)-{\hat{\tau }}_{12}\right)/{\hat{\alpha }}_{12}$;

#### 4.1.1. Evaluation Metric

In the following, different clock relation model approximations and their associated CDAs are presented. Each scheme is evaluated using time difference metric, which is defined as
(24)$$\mathrm{Time}\;\mathrm{Difference}=e≜C_{1}\left(t\right)-{\hat{R}}^{-1}\left(C_{2}\left(t\right)\right),$$
where ${\hat{R}}^{-1}\left(\cdot \right)$ is the most recent model estimate. For the linear time relation model in Equation ([Disp-formula FD15a-sensors-20-05928]), the expectation of the time difference is given by
(25)$$E\left\{e\right\}=\left(1-\frac{\alpha_{12}}{E\{{\hat{\alpha }}_{12}\}}\right)C_{1}\left(t\right)-E\left\{\frac{\tau_{12}-{\hat{\tau }}_{12}}{{\hat{\alpha }}_{12}}\right\},$$
for $t\geq t_{k}$, and where the latest time-offset estimate is calculated at time $t_{k}$. Therefore, when the clock skew estimator is unbiased so that $E\left\{{\hat{\alpha }}_{12}\right\}=\alpha_{12}$, the time difference *e* is defined by the time-offset estimation bias, and in the following the bias is used as the evaluation metric.

#### 4.1.2. Evaluation Data

The performances of CDAs for different clock relation models are evaluated using simulated data for comparative fairness. As discussed by Phan et al. [66], different data collection strategies yield different results, which may favor some models. The fairness in CDA comparison strategies is important, and a reliable result can only be achieved by using the same data for the purpose [67]. Also, the assumed model complexity of order 2 polynomial is in accordance with the reported best fit complexity (see, for example, [67]). Consequently, we use the simulated time series given in Section 3.2.3 to compare the outcome of each model.

### 4.2. Offset-Only Estimation

Offset-only model is given in Equation (Equation 17), and the corrected time for this case is given by
(26)$${\hat{R}}_{1}^{-1}\left(y\right)=y-{\hat{\tau }}_{12},$$
where ${\hat{\tau }}_{12}$ is an estimate of the parameter $\tau_{12}$. This model has only one parameter $\tau_{12}$, of which the estimate at a time instant $t_{k}$ is
(27)$${\hat{\tau }}_{12}\left(t_{k}\right)=C_{2}\left(t_{k}\right)-C_{1}\left(t_{k}\right).$$

Suppose that the offset is estimated at $t_{k}$. At a time instant $t\geq t_{k}$, the bias of the model estimation is
(28)$$b_{\tau }\left(t\right)≜E\left\{{\hat{\tau }}_{12}\left(t_{k}\right)-\tau_{12}\right\}=\left(\alpha_{12}-1\right)C_{1}\left(t\right),$$
and it is growing as the time reports of $C_{1}\left(t\right)$ progresses. The impact of the bias is a growing estimation error, which can be kept limited by a frequent estimation of the time-offset. Let us denote the time difference between successive offset estimations by Δ seconds (The time difference between two successive offset estimates Δ is also referred to as *synchronization period*, which is the time difference between two consecutive synchronization messages.). The variation of the offset compensated time reports of the clock $C_{2}\left(t\right)$ at *t* seconds after the *k*^th^ offset estimation, which is $C_{1}\left(k\Delta +t\right)-C_{2}\left(k\Delta +t\right)-{\hat{\tau }}_{12}\left(t_{k}\right)$, is shown in Figure 7 for different synchronization periods Δ. As the synchronization period increases the time error increases, and its bias is visible also in the smallest Δ value. Therefore, in case a high granularity and low-power synchronization is desired, an offset-only estimator for time synchronization is not suitable.

Suppose that there are *N* synchronization messages for estimating the time-offset. If the system permits measuring and storing two-way message exchanges (for example, using the messaging protocols in Section 5.2.1), the impact of clock skew can be kept lower than a threshold. Although optimal estimates under Gaussian random variations is the average of *N* measurements, when the delays are exponentially distributed, the maximum-likelihood estimate of the clock offset under symmetric delays has been proven by Jeske [52] to be the difference of the minimum time measurements. This result was later improved by Lee et al. [53] by employing bootstrap bias correction, and by Rhee et al. [54] using a *Particle Filter*. These methods cannot be easily generalized to other messaging schemes, and therefore receive no further elaboration.

### 4.3. Joint Batch Estimation of Offset and Skew

The clock relation model given in Equation (Equation 14) is a linear model of its parameters $\left\{\alpha_{12},\tau_{12}\right\}$, which can be estimated using the well known linear regression methods (see, e.g.,  [51]). If a table of time records [55] are used as a batch of measurements, the CDA is a batch least squares estimator. The performance of this estimator depends on the accuracy of the statistical model of the random delay terms $\varepsilon_{12}\left(t\right)$ and $\delta_{12}\left(t\right)$. If the random message delivery delay $\delta_{12}\left(t\right)$ is a zero-mean process with a finite variance and oscillator induced noise $\varepsilon_{12}\left(t\right)$ is ignored, then the estimates of the parameters are given by
(29a)$$\begin{array}{cc} {\hat{\alpha }}_{12} & =\frac{\sum_{n=1}^{N}\left(x_{n}-\frac{1}{N}\sum_{k=1}^{N}x_{k}\right)\left(y_{n}-\frac{1}{N}\sum_{k=1}^{N}y_{k}\right)}{\sum_{n=1}^{N}{\left(x_{n}-\frac{1}{N}\sum_{k=1}^{N}x_{k}\right)}^{2}}, \end{array}$$
(29b)$$\begin{array}{cc} {\hat{\tau }}_{12} & =\frac{1}{N}\sum_{n=1}^{N}y_{n}-{\hat{\alpha }}_{12}\frac{1}{N}\sum_{n=1}^{N}x_{n}, \end{array}$$
where *n* denotes the ordered index of the time values in the regression table (of *N* entries), $x_{n}$ is the time reports of the reference clock $C_{1}\left(t\right)$, and $y_{n}$ are the time reports of the local clock $C_{2}\left(t\right)$.

The batch least squares based CDA is used by Maróti et al. [56], and other works e.g.,  [57,58,59] by assuming that the random delay $\delta_{12}\left(t\right)$ is Gaussian and its samples in different messages are uncorrelated. Under these assumptions, the least squares estimator is unbiased, efficient and consistent Section 3.B in [68], which are the required properties from an estimator to attribute it as a “good estimator”. On the other hand, if the random delay $\delta_{12}\left(t\right)$ has a distribution other than Gaussian, but with a known density, the maximum likelihood estimate is usually sought. For example, the works [7,60] study exponentially distributed message delivery delay, and the model parameters are estimated using the maximum likelihood estimator.

A recursive least square estimator can also be used to estimate the model parameters as has been derived in [50]. However, this estimator uses all the past measurements, and when the skew changes in accordance with the ignored frequency drift term, its estimates quickly deviate from the true values. This problem is also observed in the batch least squares estimator since the frequency drift breaks the integrity of the regression table as elaborated by Mahmood and Jäntti [69]. When the time spacing between the entries of the regression table increases, the first order time relation model assumption fails. Solutions to this problem are to decrease the table size and/or to decrease the time spacing between the entries. However, these solutions are not always desirable or applicable due to the energy constraints or due to the required level of the time synchronization accuracy. In multi-hop networks, frequent estimation of the clock skew may also degrade the performance as elaborated on by Phan et al. [66]. This problem can be avoided by changing the time duration between the clock skew estimate updates.

The time relation model in Equation (Equation 18) ignores the time uncertainty induced by the oscillator imperfections, including time correlations and frequency drift. As we have elaborated on in the previous section, these error sources increase as the time progresses, and degrade the performance of the least square estimator. The variation of the time difference between the compensated time reports of a local clock and a reference clock is shown in Figure 8 for different synchronization periods Δ. Compared to the result in Figure 7, the average time error is decreased at the cost of estimating the clock skew parameter. This time, the estimation bias is much smaller and can be controlled by selecting Δ and the regression tables size *N* according to the required level of accuracy. However, as the time spacing between table entries Δ increases, it can be observed from the figure that the estimation bias increases. This bias is due to two factors: the neglected frequency drift term and the time report correlations due to oscillator-induced error sources.

### 4.4. Adaptive Clock Skew Estimation

The performance of the batch least square estimator can be improved by including the impact of the ignored frequency drift term in Equation (Equation 1). The effect of the frequency drift $\omega_{0}$ can be investigated by examining the bias of the estimates. Let us denote the relational frequency drift (The relational frequency drift is the parameter of the second order term of the clock relation model. This parameter can be derived by solving the equation Equation (Equation 1) for *t*, and then finding the relation of two clocks.) by $\omega_{12}$. Then, the bias in the time-offset estimate is given by
(30)$$b_{\tau }=\frac{1}{N}\sum_{n=1}^{N}\left(\omega_{12}x_{n}^{2}+\left(\alpha_{12}-{\hat{\alpha }}_{12}\right)x_{n}\right).$$

The importance of the second order parameter $\omega_{12}$ is more significant when the clock offset and clock skew estimation period Δ is high so that the time offset estimation bias grows to a significant value. Therefore, if Δ is significantly high, the synchronization accuracy can be improved by estimating the frequency drift term $\omega_{12}$.

One approach to take into account the frequency drift is to consider a dynamical model for the clock skew $\alpha_{12}$ as proposed by Hamilton et al. [61]. In the work, the authors assume a first-order autoregressive model for the clock skew,
(31)$$\alpha_{12}\left(t_{k+1}\right)=\rho \alpha_{12}\left(t_{k}\right)+\eta_{\alpha }\left(t_{k}\right),$$
where ρ is the model parameter which is less than and close to one, and $\eta_{\alpha }$ is the process noise. The skew dynamics are also studied by Yang et al. in [62,63]. After studying widely used clock skew models, the authors conclude that the skew either follows the differential equation in Equation (Equation 20), the discretized version of which reads as
(32)$$\alpha_{12}\left(t_{k+1}\right)=\alpha_{12}\left(t_{k}\right)+\omega_{12}\left(t_{k+1}-t_{k}\right)+{\tilde{\eta }}_{\alpha }\left(t_{k}\right),$$
where ${\tilde{\eta }}_{\alpha }$ is the process noise, or the constant model with $\omega_{12}\equiv 0$,
(33)$$\alpha_{12}\left(t_{k+1}\right)=\alpha_{12}\left(t_{k}\right)+{\bar{\eta }}_{\alpha }\left(t_{k}\right).$$

For the selected skew dynamics, the measurement model in Equation (Equation 19) is used for estimating the skew. In [61], the Kalman Filter (Here we do not give a summary of the Kalman filter, which can be found in standard text books on statistical signal processing (see, e.g.,  [70]).) is used to estimate the clock skew. In [62,63] several Kalman filters run simultaneously for the same purpose.

The incremental model in Equation (Equation 19) is only a function of the clock skew parameter $\alpha_{12}$ so that it can only be used for estimating $\alpha_{12}$. However, it is still required to estimate the time-offset parameter $\tau_{12}$ in order to calculate the corrected local time. One approach is to estimate it using Equation (Equation 27), and then to use these estimates to correct the output of the clock $C_{2}\left(t\right)$ by
(34)$${\hat{R}}_{3}^{-1}\left(y\right)=\left(y-{\hat{\tau }}_{12}\right)/{\hat{\alpha }}_{12}.$$

The model complexity is one of the parameters that can be tested upon to reach an optimal estimation performance. In order to reach an optimal synchronization accuracy, Kim et al. [64] have studied higher order autoregressive models for clock skew, where they have also validated the model order with well-known model selection methods. Once such a model is determined, a Bayesian optimal filter, a Kalman filter, is used for estimating the clock relation model parameters. The same line of reasoning has motived Masood et al. [65] to study alternative models with both open-loop and feedback terms. After validating a highly complicated model with the model selection procedure, they have used the steady-state Kalman Filter.

An alternative to the Bayesian estimation is the control theoretical approach, where the synchronization problem is cast as a closed loop control problem. The first work to study a control theoretical clock disciple algorithm is by Ren et al. [71]. Motivated by the fact that the NTP uses a *phase-locked loop* (PLL) type clock discipline algorithm, the authors proposed a PI controller followed by a software oscillator. This approach is further improved by Chen et al. [72] by removing the software oscillator in the loop, and introducing a strategy for selecting the gains of the controller. The method introduced by Yıldırım et al. [73] uses a PI controller with adaptive gains. This approach has certain advantages compared to the batch least square based clock synchronization. Another control theoretical solution based on low-frequency oscillators is FLOPSYNC-QACS by Terraneo et al. [74]. The authors present a control scheme that can be used when the quantization errors dominate the other time errors. Therefore, low-frequency clocks can be used for low-power time management, but they require one to take into account the required level of time resolution and quantization error into account.

In the work by Liu et al. [75], it is shown that synchronization based on a PI controller is equivalent to Kalman Filtering. Since the Kalman Filter can be written as a maximum likelihood estimate of the state with respect to the innovation, these methods are the same, in spirit, to the recursive estimator described in the next subsection.

### 4.5. Clock Skew Estimation Using Incremental Linear Model with Delivery Delay and Oscillator-Induced Correlation

In this subsection, an incremental linear model with delivery delay and oscillator-induced correlation introduced in Section 3.3.5 is used for developing batch, recursive, and numerically stable and recursive clock skew estimators.

#### 4.5.1. Batch Estimation

Linear model estimators based on linear regression have an optimal performance when the noise process is Gaussian and the measurements are uncorrelated. However, in Section 3.2, it is shown that the oscillator-induced noise is correlated, limiting the performance of the least squares estimator based on the progressive time model. This problem does not exist for the incremental linear model in Equation (Equation 21).

Let us suppose that the regression table of the time reports is composed of *N* time records. Then, the least squares estimate of the clock skew $\alpha_{12}$ for the incremental time relation model in Equation (Equation 21), and a simple estimate of the time-offset is given by
(35a)$$\begin{array}{cc} {\hat{\alpha }}_{12} & =\frac{\sum_{n=1}^{N-1}\left(x_{n+1}-x_{n}\right)\left(y_{n+1}-y_{n}\right)}{\sum_{n=1}^{N-1}{\left(x_{n+1}-x_{n}\right)}^{2}}, \end{array}$$
(35b)$$\begin{array}{cc} {\hat{\tau }}_{12} & =y_{1}-{\hat{\alpha }}_{12}x_{1}, \end{array}$$
where $x_{n}$ is the time reports of the reference clock $C_{1}\left(t\right)$, and $y_{n}$ are the corresponding time reports of the local clock $C_{2}\left(t\right)$. The variation of the time difference between the compensated time reports of the local clock and the reference clock when the local time is corrected with the estimates of the model parameters in Equation (35a,b) is shown in Figure 9a for different synchronization periods Δ. Compared to the result in Figure 8, the synchronization accuracy is improved, and the computational requirements are relaxed. Also, the time error bias is decreased.

In case the time report entries of the table are not equally spaced, the model in Equation (Equation 22) should be used. For this model, the *N* measurement *maximum likelihood estimate* (MLE) of the clock skew is
(36)$${\hat{\alpha }}_{12}=\frac{\sum_{n=1}^{N-1}x_{n+1}-x_{n}}{S(N)},$$
where we have defined the sum term as
(37)$$S\left(N\right)≜\sum_{n=1}^{N-1}\frac{{\left(x_{n+1}-x_{n}\right)}^{2}}{y_{n+1}-y_{n}}.$$

#### 4.5.2. Recursive Estimation

The clock skew estimate in Equation (Equation 36) assumes a table of time reports of *N* entries. If it is desired to recursively update the estimate when (N+1)^th^ time records are available, it is shown in [50] that the estimate is updated with
(38)$${\hat{\alpha }}_{12}^{+}={\hat{\alpha }}_{12}+\frac{x_{N+1}-x_{N}}{S(N)}\left(1-{\hat{\alpha }}_{12}\frac{x_{N+1}-x_{N}}{y_{N+1}-y_{N}}\right),$$
where ${\hat{\alpha }}_{12}^{+}$ is the updated estimate. This recursive estimator requires accumulating only one sum S(N), while estimation itself requires only a few floating point operations. Therefore, memory and computational requirements of this estimator are very low.

The performance of the recursive estimator is defined by the ignored frequency drift term, and the performance degrades drastically as the number of estimates increases. This degradation is due to the induced bias, and it causes an increasing estimation error variance as shown in Figure 9b. A solution to this problem is to give more weights to recent measurements, and exponentially dampen the old ones. The weighted version of the recursive estimate in Equation (Equation 38) is given by
(39a)$$\begin{array}{cc} \; & \Phi \left(N\right)≜\sum_{n=1}^{N-1}\lambda^{N-1-n}\frac{{\left(x_{n+1}-x_{n}\right)}^{2}}{y_{n+1}-y_{n}} \end{array}$$
(39b)$$\begin{array}{cc} \; & \Phi \left(N+1\right)=\lambda \Phi \left(N\right)+\frac{{\left(x_{N+1}-x_{N}\right)}^{2}}{y_{N+1}-y_{N}}, \end{array}$$
(39c)$$\begin{array}{cc} \; & {\hat{\alpha }}_{12}^{+}={\hat{\alpha }}_{12}+\frac{x_{N+1}-x_{N}}{\Phi (N+1)}\left(1-{\hat{\alpha }}_{12}\frac{x_{N+1}-x_{N}}{y_{N+1}-y_{N}}\right), \end{array}$$
where 0<λ≤1 is the selected weight. The variation of the time difference between the corrected local clock and the actual reference clock values in microseconds is shown in Figure 10. The depicted result clearly shows that the weighted recursive clock skew estimator outperforms all other estimators, while preserving the computational advantages of the recursive estimator.

For the clock skew estimators based on the incremental model, the clock offset can be estimated using Equation (35b). Consequently, the weighted clock skew estimator in Equation (39a–c) along with Equation (35b) constitutes the best CDA algorithm when the time reports are conveyed using one-way message dissemination. Therefore, whenever possible it should be used as the preferred CDA. Although not available, its adaptive (Bayesian) version can be derived to cover the environments that have highly dynamic temperature variations.

#### 4.5.3. Numerically Stable Recursive Estimation

A perfect clock has a time offset of $\theta_{0}=0$ s, a frequency offset of $\gamma_{0}=0$, and frequency drift of $\omega_{0}=0$. The non-zero values of some or all of these parameters are compensated by the means of a CDA. In this regard, the quality of a clock is defined by the statistical properties of these parameters. For example, the frequency offset of an acceptable quality clock must stay within a close neighborhood of 0. Correspondingly, the parameters of the clock relation model in Equation (Equation 5) should have stable values; the clock skew parameter $\alpha_{12}$ must stay within a close neighborhood of 1. One very important implication of this observation is on the numerical stability of the CDA algorithms. The round-off errors (see e.g., [76]) in the numerical representation of the time records and the clock skew estimate change the estimation error performance.

In order to demonstrate the impact of the round-off error, the batch estimator of the incremental time relation model in Equation (35a,b) and its recursive version in Equation (39a–c) are implemented using 32-bit and 64-bit floating point numbers, and the time difference between the estimated time record and the actual time in microseconds is shown in Figure 11. As it can be observed from Figure 11a, the round-off errors drastically degrade the performance of both recursive and batch estimators.

The impact of the numerical imprecision of the clock skew estimate can be mitigated by performing the numerical calculations using larger numbers. For example, one option is to define a scaled clock skew parameter β as
(40)$$\beta_{12}≜K\left(\alpha_{12}-1\right),$$
where K≫1 is a scalar yielding a large numerical representation. The value of *K* is selected based on the expected variation of clock skew between synchronization periods. If the environment is stable, the clock skew α stays in a close neighborhood of 1, and *K* is selected large if the CDA is required to react to small changes. In a more unstable environment, a smaller *K* value may be used. For example, $K=1\cdot 10^{6}$ can be used to cover most of the cases encountered in practice. Then, the recursive estimate given in Equation (39a–c) can be written as
(41)$$\begin{array}{cc} {\hat{\beta }}_{12}^{+} & ={\hat{\beta }}_{12}\lambda \frac{\Phi (N)}{\Phi (N+1)}\\ \; & +K\frac{x_{N+1}-x_{N}}{\Phi (N+1)}\left(1-\frac{x_{N+1}-x_{N}}{y_{N+1}-y_{N}}\right). \end{array}$$
On the other hand, the numerical error when calculating the time-offset estimate in Equation (35b) yields a growing estimation error variance as can be observed in Figure 11a. This is due to the multiplication of a floating point number with a higher reading of the local clock report, which causes an accumulation of the error toward the exponent of the floating point number. This can be prevented by properly defining the exponents of both of the numbers and performing the multiplication using the scaled representations.

The numerical problems discussed above can be mitigated so that the performance of the CDA can be kept comparable to the performance of the high precision implementations. The performance of the modified recursive algorithm using the clock skew estimator in Equation (Equation 41) and the truncated numbers for floating point multiplication in Equation (35b) is shown in Figure 11. The performance of the numerically improved estimator and the high precision implementation are very close to one another.

### 4.6. Results and Discussion

In this subsection, the results of numerical examples presented throughout the section are given, and their implications are discussed.

#### 4.6.1. Numerical Results

The key statistics of the results presented in Figure 7, Figure 8, Figure 9 and Figure 10 are shown in Table 5. These statistics are selected to show *accuracy* (first moment—mean value), and *precision* (second central moment—standard deviation), and *skewness* (scaled third central moment) of the estimation error(*skewness* is a measure of symmetry of the error around its mean implying that the synchronizing clock’s rate is not always higher or lower than the reference clock’s rate.). These statistics clearly show the superior performance of the weighted recursive MLE estimator using the incremental model in Equation (39a–c) compared to the others. This algorithm achieves the given performance while decreasing the computation and memory requirements compared to the requirements of the widely used batch linear regression for the progressive model.

#### 4.6.2. Discussion

The CDA algorithms presented in this section are widely used in the literature, and their properties are summarized in Table 6. All algorithms update their estimates once the readings of both the reference clock $C_{1}\left(t\right)$ and the corresponding local clock $C_{2}\left(t\right)$ are available. The operation of the clock parameter estimator depends on the estimation method and the assumed model. For the batch estimators, the clock parameters can be estimated together using a table of time records as summarized in Algorithm 2. The recursive algorithms can be implemented using Algorithm 3 with some variations for the selected CDA. The skew estimator first updates its estimate, and then use it for calculating the time-offset as given in Algorithm 3 (step 4).
**Algorithm 2** Batch clock parameter estimator1:**Time record table:**  $\left\{\left\{x_{k},y_{k}\right\}=\left\{C_{1}\left(t_{k}\right),C_{2}\left(t_{k}\right)\right\}:k=1,\cdots ,N\right\}$**Input:**  $\left\{x_{N+1},y_{N+1}\right\}=\left\{C_{1}\left(t_{N+1}\right),C_{2}\left(t_{N+1}\right)\right\}$**Output:**  ${\hat{\tau }}_{12}$, ${\hat{\alpha }}_{12}$2:Update the table entries {append the new and drop the oldest entry}3:*(for incremental model)*Calculate the time difference between the table entries4:Calculate the clock-skew using the table entries5:Calculate the time-offset ${\hat{\tau }}_{12}\leftarrow \frac{1}{N}\sum_{k=1}^{N}\left\{C_{1}\left(t_{k}\right)-{\hat{\alpha }}_{12}C_{2}\left(t_{k}\right)\right\}$6:**return**${\hat{\alpha }}_{12}$ and ${\hat{\tau }}_{12}$


**Algorithm 3** Recursive clock parameter estimator
1:**Previous time records:**  $\left\{x_{N},y_{N}\right\}=\left\{C_{1}\left(t_{N}\right),C_{2}\left(t_{N}\right)\right\}$**Previous sum parameter:**  Φ**Input:**  $\left\{x_{N+1},y_{N+1}\right\}=\left\{C_{1}\left(t_{N+1}\right),C_{2}\left(t_{N+1}\right)\right\}$**Output:**  ${\hat{\tau }}_{12}$, ${\hat{\alpha }}_{12}$**Parameters:**  λ2:
Update the sum parameter
$\Phi \leftarrow \lambda \Phi +\frac{{\left(x_{N+1}-x_{N}\right)}^{2}}{y_{N+1}-y_{N}}$
3:
Update the clock skew
${\hat{\alpha }}_{12}\leftarrow {\hat{\alpha }}_{12}+\frac{x_{N+1}-x_{N}}{\Phi }\left(1-{\hat{\alpha }}_{12}\frac{x_{N+1}-x_{N}}{y_{N+1}-y_{N}}\right)$
4:
Calculate the time-offset
${\hat{\tau }}_{12}\leftarrow y_{N+1}-{\hat{\alpha }}_{12}x_{N+1}$
5:
Save the time records
$\left\{x_{N},y_{N}\right\}\leftarrow \left\{x_{N+1},y_{N+1}\right\}$
6:**return**${\hat{\alpha }}_{12}$ and ${\hat{\tau }}_{12}$



Compared to the other CDAs given in Table 6, weighted recursive skew estimator provides significant advantages at the cost of adjusting a single weight parameter λ. In order to aid the reader, a comparison table of the derived clock parameter estimators is given in Table 7, where the computational complexity is given in big-O notation O(·). The recursive estimators are appealing for their lower computational and memory requirements. An algorithm that is a-periodic can handle occasional packet drops, which is common in wireless networks. The results given in Table 5 and Table 7, and the comparison of the methods in Table 6 imply that the numerically stable estimator achieves a higher accuracy time synchronization while using a lower amount of computational resources and providing solutions for numerical problems. Therefore, it is superior to the other estimators presented in this section.

The time synchronization period is also a function of the nominal oscillator frequency as the oscillator-induced random variations’ variance increase linearly with it (cf. Equation (Equation 11)). This observation implies that it is possible to to support longer synchronization periods by using low frequency oscillator driven clocks. In addition, the power consumption of the clock components increases linearly with frequency, which enables low-power clock implementations. This fact motivated Schmid et al. [77] to propose a Virtual High-resolution Time (VHT), which simultaneously support high-resolution and low-power clock implementation. This clock is driven by two oscillators, one being low-frequency (e.g., 32 kHZ), and other high-frequency (e.g., 8 MHz). These two oscillators are driving two counters that are phase synchronized, that is, increments of the high frequency clock are aligned with the increments of the low-frequency clock. In this case, it is possible to enable high frequency clock only when it is needed, which cuts the power consumption into a fraction. This approach requires a special hardware, and does not generalize well to software only solutions.

The CDAs are inseparable component to reach a time synchronous operation of the radio frequency communication, which depends on a frequency reference at the specified band within a certain stability range. This frequency is typically generated using crystal oscillators, which provide required stability at the cost of acceptable power consumption. However, crystals are off-chip components that cannot be integrated into silicon, which in turn increases the size and cost of the radios. If a crystal-free radio to be developed, it would not only decrease the size and cost, but also decrease the energy consumption, and increase the system responsiveness. As such, crystal-free radios can be manufactured in very small form-factors, even may lead to mote-on-chip solutions. Available on-chip resonators, however, comes at the cost of low-quality in terms of time and temperature stability. Recently, researchers have started to investigate crystal free radio platforms, which maintain frequency stable operation using the network-provided time reference. In the work by Khan et al. [78], the authors investigate the network referenced frequency locked-loop, where a reference node emits periodic beacons. In a work by Suciu et al. [79], two calibration schemes are proposed to enable crystal-free IEEE 802.15.4e [80] complaint radio development. In the work, the beacon transmissions of the network are used as the reference for calibrating radio oscillators. The work by Chang et al. [81] demonstrates a crystal-free radio implementation of time-synchronized channel-hopping networks. For network assisted and crystal-free radio development control theoretical approaches summarized above are the key enablers, as investigated, in part, in [78]. Based on the fact that these closed-loop systems have the same impact as recursive Bayesian estimators as proved in [75], the recursive skew estimator in Equation (Equation 41) can be used for improving time keeping ability of crystal-free radios.

## 5. Time Synchronization Messaging

In this section, the message delivery error sources are discussed before different messaging schemes are introduced. Thereafter, various multi-hop synchronization methods are presented, and several practical problems of time synchronization messaging are discussed. A summary of the available methods is given after presenting the time synchronization for LAN and WAN networks.

### 5.1. Background

The clocks within a network are synchronized after exchanging the most recent reports over a (lossy) communication channel, using a simple connection-less messaging also known as *time synchronization protocol*. Under ideal conditions, the current time report of the system clock is recorded and instantaneously conveyed to the destination. However, in practice, there are certain computations taking place in recoding the time reports and forming a communication frame in the source side, then, a finite duration for messaging propagation over the medium, and finally, processing of the received packet and recording the local time report in the destination side. As shown in Figure 12, all these three error sources contribute to the message delivery delay $d_{12}\left(t\right)$. The impact of the stochastic messaging errors on the time error is shown in Figure 13. As can be deduced from the figure, a high accuracy time synchronization can only be achieved by limiting the messaging time variability.

The impact of messaging delays on the synchronization error can be investigated by considering the time relation model in Equation (Equation 14). If the total delivery delay $d_{12}\left(t\right)$ has the assumed form of constant and stochastic delays, the performance of the studied CDA is expected to be close to the instantiated performances in Section 4. In other words, a precise identification (and mitigation) of messaging error sources is crucial for achieving a high synchronization accuracy. Considering the definitive nature of the error sources, in the sequel, we first review the identified sources in the literature. Then, their impact on the timestamping place within the software stack is discussed.

The WSN is composed of low-cost, low-power and short-range wireless nodes, which have limited energy and computational resources. This construction, in turn, causes these networks to experience occasional node failures, and also limitations in protocols that can run on the nodes. In this sense, the time synchronization messaging schemes must be evaluated with respect to their ability to operate under occasional node failures, and their computation requirements. Furthermore, since these nodes have limited range, the information is conveyed over multiple hops. Although supporting multi-hop messaging is important in its own right, it causes certain practical problems including defining reference clock source, deciding upon synchronization period and tackling with the security issues. In the following, we provide a comprehensive overview of the available solutions on each of these, and evaluate the presented approaches with respect to their energy and computation requirements, and their dependence on static topology.

#### 5.1.1. Messaging Error Sources

In general, the message delivery is composed of both deterministic and stochastic errors, and the second can be kept lower than a certain limit. The messaging protocols are mainly designed to lessen the negative impact of the message delivery error, which depends on a number of sources.

Chronologically, the most general error sources have been identified by Elson et al. [55] which consists of four error components:(i)*send time*,(ii)*access time*,(iii)*propagation time*, and(iv)*receive time*.

*Send time* is the time required to assemble a message in the application layer and send it to the *Medium Access Control* (MAC) layer. This delay results from the kernel processing, application induced delays, and variable delays due to the operating system scheduler. The access time is the variable delay compelled by the MAC protocols due to their propagation medium access policies. The propagation time is the time required for a message to travel from the transmitter to the receiver and it is generally negligible compared to the other delays. Finally, the receive time is the time required to process the received message and send it to the application layer. It is to be noted that the access time is included in the transmitter side delays in Figure 12.

The list above has been extended by Ganeriwal et al. [82] by adding *transmission* and *reception* times. The transmission time indicates the time required for transmitting the message bit by bit in the physical layer. The reception time refers to the time needed for receiving a message in the physical layer and passing it to the MAC layer. The transmission time is a frame length dependent delay, and it follows the access time in the transmitter side. The reception time is the first delay of the frame during the reception. These two delays are equal to each other, and have deterministic nature.

The delays right before transmission and after the reception of a message have been studied by Maróti et al. [56]. They have extended the transmitter side delays with *encoding time*, which indicates the time required for encoding the current message, before the transmission time, which covers the processing delay associated with transforming the bit information to the electromagnetic waveform right after raising the transmitter side interrupt. The receiver side delays are extended to cover *decoding time*, which refers to the time needed for transforming the electromagnetic waveform into the bit information and decoding the message in the receiver side. The *byte alignment time*, which stands for the time incurs in the receiver side due to different byte alignment of the transmitter and the receiver. The *interrupt handling time* of the receiver side is the delay that radio chip waits for the processor to finish the current instruction or the current critical section before switching to the interrupt handling subroutine.

In the work by Ferrari et al. [57], the transmitter side delays have been further divided by including the *software delay* and *calibration time*. The software delay is the time required for processing unit to prompt a message transmission by accessing certain registers. This time is affected by the interface between the radio and the processing unit. The calibration time is the time required to lock to the operating frequency of the voltage controlled oscillator (VCO) before starting the transmission.

The overall error sources are tabulated in Table 8, where the transmission message formation and transmission frame preparation times are included. In the table, the induced timing error type is also specified under each error source. As can be seen, the identified error sources above are closely coupled with the hardware and the software architecture of the nodes, but most of the error sources can be kept constant, for example, by keeping the time synchronization frame length constant.

The software delays in transmission (software delay) and reception (interrupt handling time) side are random, and their variability depends on overall system design. In particular, the variability of the delays induced by these components depends on the processor load. The calibration time usually has a guaranteed delay under normal operating conditions, and has a low variability. The access time depends on the transmission medium, and its characteristics depend on MAC layer implementation. The joint effect of these random sources can be described using queuing theoretical models such as M/M/1 queue which assumes exponentially distributed delivery delay (see e.g., [7,60] and references therein). The M/M/1 queue assumes a single server with exponentially distributed service times and Poisson distributed arrivals. In practice, the software delay, interrupt handling time, and access time are all independent queues, and their joint impact may require more complex statistical models. Another approach is to invoke the central limit theorem by assuming a large number of independent uncertainty sources as it has been done by Elson et al. [55]. The same line of reasoning has motivated other researchers to assume Gaussian distributed random delays, see e.g., the works by Hamilton et al. [61], and Leng and Wu [83].

#### 5.1.2. Timestamping

Ideally, the time stamps must be stored when the emission of the message starts at the transmitter, and when the reception of the same message is detected by the receiver. This way the random error source affecting the time records are totally eliminated (cf. Table 8). However, in practice, the frames are formed well before the actual transmission starts, and buffered for efficiently handling the MAC layer rules for accessing the propagation medium. In this case, it is not possible to avoid random access time. In the same line of thinking, as the timestamps are taken further away from the physical layer, the more time errors are accumulated on the time information. Therefore, the messaging error sources can be mitigated by storing the timestamps of both transmitter and receiver sides as close as possible to the physical layer of the node.

This problem is considered by Maróti et al., and they have proposed MAC layer timestamping [56]. Later, this approach elaborated further by Cox et al. [84] and Aoun et al. [85]. Considering a general radio receiver, a start of frame is detected after decoding preamble and a certain start of frame indicator. When a receiver decodes this indicator, it can be safely assumed that the radio has received a frame regardless of its integrity or information content. For example, for IEEE 802.15.4-2006 compliant radios the start of a frame is delimited with a specific byte referred to as Start of Frame Delimiter (SFD) byte [86]. After decoding this byte, the receiver can store its local time, and if the transmitter is not using buffered design, it can append its own local timestamp to the end of the frame as shown in Figure 14. Considering the messaging error sources, for a system utilizing the MAC layer timestamps, only interrupt handling time cannot be avoided unless the hardware supports other means to store the receiver side timestamp. One example is presented by Asgarian and Najafi [87] for connectionless Bluetooth LE beacons.

It is to be noted that the error sources in Table 8 can be controlled by moving software components to hardware implementations. In this case, the software delay in the transmitter side, and interrupt handling time in the receiver side would have deterministic characteristics. The only delay source that cannot be changed is the access time. However, such a solution is not always feasible. For example, some of the IoT devices have very limited resources, and the hardware manufacturers prefer to limit the software access to certain hardware features associated with the communication timings. Similarly, some wireless communication technologies are mostly implemented in the hardware, and it is not possible to control when to capture the timestamp. Therefore, the achievable time synchronization accuracy depends on the underlying hardware and the wireless communication technology as the timestamping method is dictated by them.

### 5.2. Messaging Schemes

The information acquired on the messaging error sources depend greatly on the messaging schema utilized by the system. In the literature, there are four well-known signaling schemes: two-way message exchanges, one-way message exchanges, receiver-only synchronization, and receiver-receiver synchronization [7].

#### 5.2.1. Two-Way Message Exchanges

This messaging scheme is the most widely used mechanism for exchanging the time reports of two adjacent nodes, which are referred to as sender and receiver nodes. In this scheme, a sender node requests the most recent time report of one of its neighbors. The receiver node responds with its most recent time report along with the reception time of the request. The sender node calculates the clock relation model parameters using all four time information shown in Figure 15. Since there are two time reports for each node ($T_{1}$ and $T_{4}$ for sender, and $T_{2}$ and $T_{3}$ for receiver), this scheme can achieve very high synchronization accuracy. Furthermore, suppose that $d=d_{12}=d_{21}$, then we have
$$\begin{array}{cc} \theta & =\frac{\left(T_{2}-T_{1}\right)-\left(T_{4}-T_{3}\right)}{2},\\ d & =\frac{\left(T_{2}-T_{1}\right)+\left(T_{4}-T_{3}\right)}{2}, \end{array}$$
when the time uncertainties are ignored. Therefore, this message exchange method can be used for estimating time offset and messaging delay, which are needed to reach very high accuracy clock synchronization. In Section 4, our focus was on one-way message dissemination. The interested reader is invited to investigate [7] for different CDAs of two-way messaging. These methods do not take into account the correlation in the time records. Since all the measurements are limited to one round-trip duration, its impact is negligible.

The contemporary time synchronization solutions are built upon two-way message exchanges [10,11,12,13], and they require pairwise communication of the nodes. This, however, implies that each node must support both reception and transmission of the clock information, which may not be suitable for some of the IoT deployments. For example, if a Bluetooth sensor is periodically broadcasting some information as an advertisement, the listening devices may not initiate a bidirectional communication. Furthermore, for each synchronization, two message exchanges are required, whose energy requirement is twice as the one-way message dissemination scheme. Therefore, despite the fact that this scheme can achieve high synchronization accuracy, it has certain drawbacks that should be taken into account.

#### 5.2.2. One-Way Message Dissemination

The energy constraints of the low-power networks lead to one-way message dissemination scheme, where a root node broadcasts synchronization messages, and all of its neighbors receive it as shown in Figure 16. In this scheme, the receiver nodes synchronize to the root node using only one-way messages so that the power requirements of the time synchronization method is reduced by half.

The achieved energy saving comes at the cost of increased messaging uncertainty on the transmitter side. In two way message exchanges, the time error sources are constraint to the one messaging duration i.e., $T_{4}-T_{1}$ in Figure 15b. However, for one-way message dissemination, the time error is accumulative and must be mitigated by some other means in order to achieve higher synchronization accuracy. For example, the time stamping can be moved to as close as possible to the radio interface in the software stack, e.g., to the MAC layer.

#### 5.2.3. Receiver-only Synchronization

The high accuracy synchronization that can be achieved by two-way messaging and reduced power requirements of one-way message dissemination schemes can be combined by considering the broadcast nature of the wireless communication. One option is to allow all the nodes in a common neighborhood of two nodes that are exchanging time messages to overhear the ongoing communication as visualized in Figure 17. In this case, the receivers can utilize the acquired time information to achieve a high accuracy clock synchronization, while maintaining power requirements on the order of one-way message dissemination schemes [88].

#### 5.2.4. Receiver-receiver Synchronization

The broadcast nature of the wireless sensor network can be exploited to eliminate the transmitter side errors using receiver-receiver scheme shown in Figure 18. In this scheme, a root node first broadcasts the reference time, which is received by its neighbors. The neighbors store their local timestamps at beginning of the reception. Then, the receivers exchange the time information they have stored with each other. The transmitter side delays are eliminated since both neighbors receive the transmitted frame at the same time. Although there might be a time difference due to different propagation delays, it is negligible for short-range networks. This elimination yields better performance as the transmitter side random delays constitute the main body of the random messaging delays [55].

#### 5.2.5. Discussion

The four types of messaging schemes have different implications on time synchronization accuracy, power consumption and network topology. In this regard, the one-way message dissemination requires the smallest number of transmissions in the network, and does not dictate a topology at the cost of controllable degradation (through appropriate timestamping method) in synchronization accuracy. The two-way message exchanges provide the highest amount of information on the clock parameters, whereas it requires two message exchanges for each synchronization period, which implies higher energy and computational requirements, and pair-wise operation. However, it can be used in cases where the lower layers of the communication protocol cannot be modified, as has been demonstrated by Son et al. [89] over constraint application layer protocol CoAP [90] to achieve an acceptable synchronization accuracy. The receiver-receiver synchronization scheme can achieve a higher accuracy than the one-way message dissemination scheme at the cost of using higher amount of energy and computational resources. Since this scheme depends on the existence of a transmitter node, its node failure resilience is higher than two-way message exchanges but lower than one-way message dissemination. The good properties of two-way message exchanges and one-way message dissemination are combined in receiver-only synchronization. This scheme has low resilience against node failures, but has low-power requirements and yields high synchronization accuracy.

The properties of these four message exchange schemes imply that if one has a freedom to choose the messaging scheme, they can make the decision based on the required synchronization accuracy for a given energy and computational resources budget. For most of the IoT deployments, however, a system designer is obligated to choose from off-the-shelf components to integrate with the system. In this regard, another criteria that must be considered is the required level of modifications to the standard compliant-systems. For such deployments, receiver-receiver synchronization is an attractive option as it eliminates the transmitter side delays without requiring special timestamping. The receiver-only synchronization can be used if some nodes can support more resources compared to others. The two-way message exchanges are the preferred scheme when the deployment can support two-way messaging. Therefore, an IoT practitioner has to select the messaging scheme considering several practicalities, including also the multi-hop synchronization support.

### 5.3. Multi-hop Synchronization Schemes

The messaging schemes introduced in the previous subsection can be used by the nodes within the communication range of a reference (root) node. Some of the nodes in practical WSN deployments, however, are placed further away from the root node so that they can be reached out by conveying the information over several hops. With respect to the time synchronization, the reference time information of the root node is contaminated by the hopping process. The overall impact of this process depends on the hopping and adopted messaging schemes. Therefore, the multi-hop time synchronization schemes are definitive for the achievable time synchronization performance.

In general, the network-wide time synchronization is closely coupled with network routing. For example, in Routing Integrated Time Synchronization (RITS) protocol [91], the synchronization messages are piggy-backed onto the routing broadcasts. Although this method has certain benefits in decreasing the synchronization messaging overhead, it does so by limiting the performance. In the remaining part of this subsection, we do not consider an integrated approach.

Multi-hop synchronization schemes aim at providing a network-wide coherent notion of time. In the literature, the available multi-hop schemes can be divided into two classes: centralized methods, where a root node is (dynamically) elected to provide reference time, and distributed methods, where the nodes reach a consensus of common time. In the following, we first summarize two centralized schemes: cluster-based receiver-receiver and spanning-tree based schemes. Then, a diffusion based scheme, which uses spatial averaging, is presented. Afterwards, distributed and local gradient-based scheme is summarized. Finally, an overview of consensus-based schemes is given.

#### 5.3.1. Cluster-Based Synchronization

The receiver-receiver messaging scheme introduced in the previous section enables a very simple multi-hop scheme [55]. Consider the messaging scheme in Figure 18a, and now suppose that there is another root node outside of the communication range of the first root node as visualized in Figure 19a. All the neighbors of the first root node, say nodes 1 and 4, can translate the time information of one to the time information of the other. Similarly, the neighbors of the second root node, say nodes 4 and 7, can translate to each other’s time information. Now, if node 1 relays certain information with its local timestamp to the node 7, the node 4 can translate the timestamp first to its local time, and then to the time of node 7. This multi-hop scheme allows very simple network-wide time synchronization by forming *clusters* of synchronized nodes around *gateway* nodes as illustrated in Figure 19b.

The main disadvantage of this scheme is due to its dependence on the receiver-receiver messaging scheme. This scheme requires a large number of data exchanges between all the neighboring nodes, which causes increased power consumption, and poor scalability. In the work by Jain and Sharma, these requirements are restrained by controlling the transmission power [92], which in turn dictates the size of each cluster. On the other hand, this multi-hop scheme has a strong dependence on the availability of the gateway nodes, making it sensitive to single node failures.

The scalability and failure sensitivity problems of this method are addressed in the work by Palchaudhuri et al. [93]. In the modified method, all the nodes broadcast reference time information after receiving from the neighbors. This way, the time synchronization is flooded to the entire network by forming a spanning tree. The authors discuss that the overhead associated with flooding of the receiver-receiver messaging scheme can be overcome by broadcasting the reference time when a receiver requests it.

The main advantage of receiver-receiver messaging based multi-hop synchronization lies in its ability to achieve high synchronization accuracy even when the hardware platform cannot be modified to mitigate the messaging error sources [94]. Although the receiver-receiver messaging scheme eliminates the transmitter side errors at the cost of increased communication overhead, Coefficient Exchange Synchronization Protocol (CESP) [94] offers an optimized solution that can be used in different IoT deployments. A similar approach is presented in the work by Sridhar et al. [95], where low-power Bluetooth LE beacons are used to transmit time information to high-end receivers. In the proposed scheme, the high-end receiver is used for translating the time information of each receiver’s time information.

The general IoT deployments shown in Figure 1 makes it difficult to implement the receiver-receiver scheme since all the nodes must exchange the time information they have acquired during the latest synchronization round. A mixed, and optimal, approach has been proposed by Kim et al. [96], where they integrate the time data exchanges between high-end transmitter node and low-power nodes with the sensor data exchanges. Although the high-end transmitter makes use of two-way messaging to estimate the clock offset and messaging delays, the receiver uses one-way message exchanges to estimate the skew. This hybrid strategy is optimal in the cases two-directional communication is possible. The design assumption of having resourceful gateways well suits to IoT deployments and this approach is expected to be of practical use.

#### 5.3.2. Spanning-Tree Based Synchronization

The complexity associated with the cluster-based receiver-receiver multi-hop synchronization has led researches to investigate alternative schemes with less overhead. In the work by van Greunen and Rabaey [97], a hierarchical tree of the nodes is formed in order to achieve time synchronous network operation. In this method, a root node, which may be re-elected every time the synchronization protocol runs, initiates the synchronization procedure by synchronizing to its neighbors within the communication range. The nodes synchronized to the root node, then, synchronizes to their children nodes, which could not receive the root node’s synchronization messages. The process continues until all the nodes are synchronized to the root node.

The process described above floods the reference time information of the root node throughout the network by forming a spanning-tree of the hierarchical relations of the nodes. The root node is at level 0 of the tree, and its children are at level 1 as depicted in Figure 20. The flooding provides an easy way to achieve a synchronized network, regardless of the messaging scheme utilized to exchange the time information. The Tiny-Sync and Mini-Sync [98], the Lightweight Time Synchronization (LTS) [97], and the Timing-sync Protocol for Sensor Networks (TPSN) [82] use two-way message exchanges; the Spanning Tree-based Energy-efficient Time Synchronization (STETS) [99], its improved version R-Sync [37] and self-recovering version Self-Recoverable Time Synchronization (SRTS) [100] use receiver-only scheme; and the Flooding Time Synchronization Protocol (FTSP) [56], which is the de-facto synchronization protocol in WSN literature, uses one-way message dissemination.

The flooding of synchronization messages within the spanning-tree requires certain practical adjustments in order to prevent traffic congestion and frequent packet losses in the network. These adjustments include also delaying the transmission of the reference time from lower levels to the higher ones, which increases the flooding duration. Due to the differences in the estimated clock skew of the nodes in a branch, as the level increases the estimated skew error increases, and the time error increases linearly with the number of hops [58].

The increase in the time error with the network diameter problem can be overcome by using faster flooding approaches. One option is to allow packet collision by exploiting *the power capture effect* [101]. Another is to exploit constructive interference that can be obtained by precisely programming the forwarding instance of the same packet [57]. Although these methods allow mitigation of the increase in the timing error, the rapid flooding provided by PulseSync [58] offers a significant improvement in multi-hop time synchronization accuracy by compensating for the accumulated time error. An alternative solution has been introduced by Wang et al. [102], where the nodes adjust their clock values before updating their clock parameter estimates.

The solution provided by the PulseSync protocol is limited in the sense that some practical networking problems still may degrade its performance [59]. First, the neighbor contention may slow down the flooding speed. Second, a precise network schedule is required to prevent packet collisions and neighbor contention. And lastly, reliable rapid flooding is problematic in wireless communications due to frequent packet losses. These issues can be efficiently handled by synchronizing the clock speed of each node in the network along with their time. The solution provided by Yıldırım and Kantarcı in [59] combines the benefits of distributed gradient algorithms and flooding based synchronization methods.

The spanning-tree based synchronization has several problems due to the distributed nature of the WSN deployments. In particular, location dependence of temperature and supply voltage level (This may be due to poor connectivity as nodes with poor connectivity may try several re-transmissions, which depletes their battery faster than the battery of the other peers in the network.) yield different clock rates in distant regions of the network. Thus, integrating time information flooding and routing in such deployments should be discouraged as discussed by Schmid et al. [103]. The network-wide clock speed agreement [59] also provide robustness against inhomogeneous temperature and supply level distributions within the network.

The control theoretical clock synchronization proposed by Chen et al. [72], and FloodPISync protocol proposed by Yıldırım et al. [73] are spanning-tree based multi-hop scheme. These methods provide improved performance due to their lower RMS time error compared to the least-square based CDA.

The nodes in an IoT deployment are likely to have heterogeneous capabilities in terms of computational and communication resources. Some nodes may have enough resources to utilize synchronization schemes for LAN networks (e.g., NTP, SNTP or PTP), while some others may have just enough resources to overhear the synchronization messages among the high-end nodes. This approach not only enables time synchronous operation among heterogeneous nodes, but also provide significant power saving. One way to implement this idea is to use the receiver-only messaging scheme, where the high-end nodes perform two-way time information exchanges, while the low-end nodes overhear these packets. The Group-wise Pair selection Algorithm (GMA) is a multi-hop scheme presented by Noh et al. [104] and STETS [99] enable such deployments, but suffer from isolated nodes and sensitivity to group leaders problems. A solution to the first problem is provided in R-Sync [37], and a solution to the second is provided in SRTS [100]. An interesting approach would be to implement a synchronization scheme, where low-end nodes can overhear PTP [13] messages among high-end nodes. Although this idea has not been demonstrated in the literature, it has a potential to provide a high-accuracy synchronization with minimal development time and energy demands.

Some industrial IoT deployments are naturally distributed in different topologies. For example, in underground mining, there is a network of base stations in tree topology, and sensor nodes are connected to a single base station in start topology [105]. In this case, different synchronization strategies can be employed for a given synchronization requirement and energy budget of an application. Although a simple offset-only CDA is used in [105], the problem can be solved using different approaches. One option is to use PTP for base stations, and utilize receiver-only synchronization among the sensor nodes.

#### 5.3.3. Synchronous Diffusion

The centralized nature of the multi-hop schemes presented above can be relaxed by performing spatial averaging of the received time information from the neighbors using the Time Diffusion Protocol (TDP), first introduced by Su and Akyildiz [106]. In this approach, dynamically elected master nodes broadcast time information to their neighbors. The elected neighbors, which are called diffusion leaders, acknowledge the received messages. The received acknowledgment messages are used for determining next diffusion time by calculating the time difference between message transmission and acknowledge reception. All the diffusion leader nodes also broadcast their diffuse time information to their children. The elected diffusion leader nodes repeat the procedure until all the nodes are synchronized. This method is self-configuring, takes into account energy constraints of typical WSN deployments, and resilient to node failure, adverse communication channel, and to the node mobility [106].

The election of the master nodes and diffusing the time information throughout the network limit the applicability of TDP. An asynchronous distributed diffusion method is introduced in a work by Li and Rus [107]. The method synchronizes to the average time of all the clocks in the network.

#### 5.3.4. Distributed Synchronization

In the work by Solis et al. [108], it is shown that the sum of relative time offset between nodes in a closed loop should sum to zero. This observation leads to an algorithm with a very simple interpretation: each node updates it local time by averaging all of its neighbors’ time-offset estimates. The averaging is also referred to as spatial smoothing, and it is limited to the one-hop neighborhood of a node. This method is very similar, in spirit, to the distributed diffusion based method of Li and Rus [107]. Similarly, Yıldırım et al. [73] have proposed AvgPISync protocol, which calculates the control rules using averaging. Spatial averaging does not rely on prior topological knowledge, or existence of certain neighbors, that is, it is resilient to node failures and topology changes. Therefore, distributed algorithms make use of collective information to reach to a common notion of time, as we summarize next.

*Gradient clock synchronization:* The *global* time synchronization is to reach a common notion of time within the network. A straightforward solution to this problem is to extend the pair-wise synchronization methods to synchronize the associated nodes of each hop. It can be easily seen that such an extension implies a centralized structure, where the root node is at the center. All other nodes are residing on a branch of the tree. However, this hierarchical node relationships disregard the *locality* of the clock skew [109], which can be exploited to reach to the spatial *gradient* property of the clock synchronization algorithms [110].

As discussed previously, the frequency drift is a local phenomenon, and so is the clock skew. When one considers the pair-wise synchronization, this information implies that as the number of hopes from the root node increases, the achievable synchronization accuracy decreases due to gradient of the clock skew. This implies that if it is left unattended, the clock skew increases with the network diameter. The synchronization algorithms allowing such a behavior are said to have the gradient property, which has been first introduced by Fan and Lynch [110]. They have shown that the clock skew of the algorithms satisfying the gradient property is bounded from below. The tightness of this bound can be discussed with respect to an upper bound as has been done by Locher and Wattenhofer [111].

The theoretical results for gradient clock synchronization [110,111] have been used by Sommer and Wattenhofer [109] to develop a practical time synchronization algorithm for WSN. In this approach, a logical clock is defined which updates its time output at a specific rate. All the nodes in the network strive to agree upon the current logical time. The nodes periodically broadcast messages of logical clock value and clock rate, and they use all the messages they have received from all their neighbors to update current logical clock rate and offset. A more elaborated version of the same approach is presented in a work by Pinho et al. [112]. The gradient clock synchronization is localized in nature, and it is resilient to node failures or topological variations since it is a distributed method.

*Consensus-based synchronization:* The gradient synchronization methods are local in spirit. A natural question is whether a network-wide, convergent and distributed synchronization algorithm can be developed. The answer to this question is not only positive [113], but also it can be shown that all the nodes in a network converge to the same clock if certain conditions are met [114]. In these network-wide solutions, the nodes agree upon both clock rate and time of logical clocks. This agreement is referred to as *consensus* in distributed systems [115].

The Average TimeSync (ATS) consensus-based solution presented in [113,114] converges to a clock rate and a clock offset using a weighted average from all the nodes in the network. This solution has a higher communication demands than the gradient clock synchronization. An approach that updates its clock parameters using the information received from its single-hop neighbors has been proposed by Maggs et al. [116]. Reaching to the consensus of the average clock might be slow if the network is distributed in a large area. One solution is to synchronize to the fastest clock or to the maximum value clock as has been done by He et al. [117]. Based on this work, it is later shown that a maximum-value-based-consensus algorithm can converge with probability one and exponentially fast under bounded timing noise [118]. The performance of maximum-value and average-value -based consensus schemes are compared using extensive empirical data in [119]. Based on the findings, the authors have proposed a maximum-minimum time synchronization protocol, which is a special case of more general min-max consensus algorithms [120]. In these schemes, the clock rate adjustment is independent for each node, which may cause frequent clock rate adjustment or even divergence. This problem is addressed in the work by Sun et al. [121], where the clock rate is adjusted less frequently than the clock offset. The convergence of this approach is proven in [122]. The topological conditions of convergence of consensus-based algorithms under bounded communication delays is investigated by Tian et al. [123], and based on the divergence conditions, the least-squares based time synchronization is proposed [124]. An algorithm which takes into account the measurement noise is proposed by Stanković et al. [125], and convergent algorithms for both clock skew and time offset are presented.

The main problem of the consensus-based methods is their increased memory requirements as each node must acquire and store time information of their neighbors. This requirement creates significant problems in dense deployments as the computational, memory and communication requirements usually exceed the capabilities of low-end platforms. Another problem, known as *time partitioning* occurs when the clock rates are not synchronized [126]. This problem arises when the intermediate nodes have slower clocks compared to the ones at the edges. In this case, the nodes closer to the one edge synchronize to the one clock, whereas the nodes closer to the other edges synchronize to another. This problem can be avoided by synchronizing also the clock rates.

*Network-wide synchronicity:* The network-wide clock synchronization methods aim at providing a coherent notion of time among the sensor nodes. However, some application scenarios require a network-wide synchronism in performing certain tasks/actions rather than a coherent notion of time. Such tasks may involve synchronously sampling a spatio-temporal field, for example, for structural health monitoring [22]. The importance of network-wide synchronicity for machine-to-machine type communication networks is discussed by Bojic and Nymoen [44].

Synchronicity is a reoccurring theme in nature; for example, the flashing times of fireflies living in certain parts of southeast Asia are synchronized. The behavior of these biological systems can be modeled as pulse-coupled oscillators, which has a known mathematical formulation [127]. Due to the similarity of the periodic actions performed by the swarms with the WSNs, the aforementioned results have inspired Werner-Allen et al. [128] to introduce Reachback Firefly Algorithm (RFA), which provides a network-wide synchronicity for wireless sensor networks.

Another option to achieve network-wide synchronicity is to use black-bursts, which provide means to allocate the medium for transmission of special packets [129]. The black-bursts may be used for aligning transmissions in the network, and thus enable achieving network-wide synchronicity [130].

#### 5.3.5. Discussion

The cluster-based synchronization has the same advantages as the receiver-receiver messaging scheme and provides an option to reach a high synchronization accuracy with minimal modifications in off-the-shelf components. Spanning-tree presents itself as a natural choice for one-way message dissemination, although other messaging types can also use it. It can reach high synchronization accuracy if the hop-by-hop error accumulation is mitigated by, for example, using MAC layer timestamping. Synchronous diffusion provides a way to distribute the time information of a root node throughout the network by diffusing the time. It uses the spatial averaging and has similarities with the distributed algorithms. The distributed algorithms aim at providing network-wide synchronization while being resilient to node failures and topology variations.

The four multi-hop synchronization schemes summarized above have certain advantages compared to one another. Although for an IoT deployment, there is an almost always a root node (possibly connected to the wide-area or local-area network), the spanning-tree based multi-hop scheme along with one-way-message-dissemination-based messaging presents itself as a natural choice. However, hybrid approaches, e.g., spanning-tree and consensus, are recently emerging, and they provide robustness against the practical problems arising due to the distributed nature of the IoT deployments. Therefore, if it is possible to select a multi-hop scheme, one should make the decision based on the possibility of implementing a certain messaging scheme, the network size and the method’s scalability, and its robustness against the node failures and topology variations.

### 5.4. Practical Problems

The messaging schemes that have been presented up to this point make certain assumptions about several practicalities associated with time synchronization. In particular, they assume a certain synchronization period, a certain way of selecting the reference clock source, and a way of defending against malicious attacks. Here, we overview available solutions to these three problems.

#### 5.4.1. Synchronization Period

All messaging schemes aim at providing a high accuracy synchronization with minimal network activity in order to consume as less energy as possible. We have shown in Section 4 that the time error increases as the synchronization period increases. On the other hand, it can be easily concluded that the amount of energy consumed by the time synchronization task is defined by the period of the synchronization messages. In other words, low-power and high granularity synchronization are an oxymoron as discussed by Schmid et al. [131]. Therefore, the trade-off between low-power and high-accuracy synchronization can be adjusted by dynamically changing the synchronization period [36].

Most of the available methods have a static synchronization period, and the designer must choose the trade-off point between high accuracy and low power operations in the design time. The operating point is decided by making certain assumptions about parameters defining the clock relation model in Equation (Equation 5). In particular, the frequency drift is the quadratic term in the estimation error bias in Equation (Equation 30), it causes the *frequency error impact* on the clock synchronization [103]. Its affect on the time error in Equation (Equation 24) can be compensated by frequent estimation of the clock skew. Thus, an adaptive algorithm can adjust the synchronization message transmission period according to the time error: the transmission rate is decreased when the time error is smaller than a threshold, and it is increased when the error increases above the threshold [36].

The Rate Adaptive Synchronization (RATS) alters the synchronization period as a function of the time error [132]. A similar approach is used in dynamic FTSP (D-FTSP) where the transmission interval of the broadcast messages are altered based on the variation of the clock skew estimates [133]. For time-synchronized channel-hopping networks based on IEEE 802.15.4e, the overall network performance in terms of reliability, delay and power consumption can be improved by adaptively changing the synchronization period [134,135].

The clock-skew changes with environmental parameters such as temperature and supply voltage variations. In Temperature Compensated Time Synchronization (TCTS), the frequency drift is estimated as a function of temperature, and the synchronization period is altered as a function of frequency drift estimate [136]. In a work by Yang et al. [63], an aided multi-model Kalman filter is used for estimating the clock-skew when the environmental conditions are stable, and when they are changing. The work aims at creating a calibration table of clock-skew versus temperature, and updates the table online when a new temperature or skew value is detected. The same line of reasoning has led Elsts et al. [137] to study the impact of temperature for relaxing re-synchronization period of time-synchronized channel-hopping networks. They achieve significant energy saving by calibrating the clock-skew variation with temperature. The impact of supply voltage on clock-skew is investigated by Jin et al. [138] for creating a calibration table to relate clock-skew with supply voltage value. However, the practical importance of supply voltage variation is limited since the voltage can be regulated with basic electronic components, and its impact can be easily kept under a certain value. Therefore, if the nodes are equipped with a temperature sensor, the clock-skew variation can be calibrated, which yields significant power savings by prolonging the synchronization period.

#### 5.4.2. Reference Clock Source

Thus far, we have implicitly assumed that the network entities aim at synchronizing to a global clock source provided by a node or an entity in the network. This approach provides means to synchronize to a network-wide time, which may or may not be in a universal time base, using time synchronization messages. However, it is possible to take an alternative approach, and equip the nodes with a specialized hardware to let them acquire universally valid reference timestamps on the same processing unit.

An obvious choice for outdoor deployments is to use a GPS signal, which provides very accurate time base information. For example, ZebraNet nodes are equipped with a GPS receiver [139], which limits the lifetime of the battery-powered nodes due to significant power consumption. A similar approach is to use a specialized hardware to receive pervasive time signals as has been done by Chen et al. [140]. The main issue with such solutions is the sensitivity of their receivers to the environmental conditions, which limits their utility in a broader class of use cases and applications. This limitation can be overcome by utilizing available ambient periodic signals to generate a stable reference time base. For example, the pilot tones generated by the FM radio stations provide a stable reference signal [141]. Since the most widespread periodic source is the main power line, it can be used as a de-facto source as has been studied by Rowe et al. for the purpose [142]. A similar approach is to use the intensity variation of the fluorescent light with the AC power source [143].

The specialized hardware can be built with very energy efficient components so that very low-power time synchronization can be achieved. However, special hardware requirement is usually avoided due to increased development, production and deployment costs. In this regard, a hybrid approach is to equip some of the nodes, say root or master nodes, with specialized hardware to generate high accuracy reference time is very attractive. This way, a high accuracy and stable clock synchronization can be achieved while keeping the costs low. For example, Gupchup et al. have proposed Phoenix [144], and Dai and Han have introduced TSync [145] as hybrid solutions.

An alternative is to use periodic transmissions of the other wireless communication technologies operating in the same frequency band. A method proposed by Hao et al. exploits periodic beacons of the Wi-Fi access points to generate stable time reference for IEEE 802.15.4 nodes operating in the same band [146]. This approach does not require any specialized hardware, but only software that acquires received signal strength measurements and a signal processing method to calculate the period of the beacons. Therefore, this is a promising method in urban areas.

The low-power sensor nodes can be deployed in numbers to acquire redundant measurements of the same physical phenomenon. Such a redundancy also provides a basis to observe the same events over different data streams. This option is investigated by Bennett et al. [147], and inertial data streams of different sensors affected by the same human actions are used for identifying the instance a specific event has happened. The authors show that these correlated time series can be used for identifying *alignment points* of time, and based on these time instances time offset and clock skew can be compensated. Later, the method is improved by including several filtering mechanisms to increase its robustness [148]. The same line of reasoning has recently motivated Shaabana and Zheng [149] to introduce a machine-learning-based identification of alignment points using a support vector classifier. Although these methods are very attractive for synchronizing offline data acquired in a small environment, and may provide improvements in signal processing techniques that require time alignment, their application is very limited in larger scale deployments.

#### 5.4.3. Security

Any wireless communication is over insecure wireless propagation medium. For these types of networks, in general, it is easy to: (i) attack the system by pretending to be another node in the network, and sabotage the performance by eavesdropping, frame replay, and spoofing; (ii) insert false data into the system; (iii) make requests with a rate larger than the system can handle, and fall into the denial-of-service state. Although these problems have known solutions for contemporary wireless systems, for low-power and short-range technologies security solutions are limited due to their energy, computation, and memory constraint [150]. In this regard, a time synchronization messaging scheme is prone to [151]: (i) manipulation of the time information by packet interception; (ii) eavesdropping the time synchronization messages to store and replay them in order to degrade the notion of time in the network; (iii) spoofing the reference clock source by pretending to be the root node.

The security issues in the time synchronization messaging protocols can be addressed by well-known methods such as authentication and encryption [152], and refusal to forward suspicious timing messages identified by traffic monitoring [153,154].

*Authentication and encryption:* In order to ensure secure and trusted network operation, the nodes must be authenticated and all the information must be encrypted. In this case, the unauthenticated nodes cannot join the network, and insecure data will be discarded by the genuine nodes of the network.

The methods suggested in [150,151] can be used for authentication. The messages can be end-to-end or hop-by-hop encrypted. In the former, only sender and receiver can decrypt the message, whereas, for the latter, the data can be encrypted by intermediate nodes as well. The end-to-end approach is less exposed to malicious nodes [151].

*Removal of suspicious packets by traffic monitoring:* Once the network is deployed and initialized, the nodes synchronize their times. As it is elaborated in Section 3, the time reports of one clock do not quickly deviate. Thus, if a node receives a reference time report that is very different from its synchronized time, the received time information can be treated as a suspicious message. The same line of reasoning can be extended to a network wide monitoring, as it has been done in [154,155]. In this case, the network statistics are collected and analyzed in order to detect a deviation that might reveal the presence of an adversary in the network.

*Secure time synchronization protocols:* The security issues presented thus far, and the provided two mechanisms have led researchers to design secure versions of the well-known protocols. The security toolbox introduced by Ganeriwal et al. [156] aims at detecting malicious attacks for two-way message exchanges and stopping time synchronization in case of an attack. They consider, in particular, the pulse-delay attacks. The security issues in flooding time synchronization has been studied by Huang et al. [157], and they have introduced different methods to identify five different attacks that may hinder time synchronization operation. The methods presented by Sun et al. [158] aim at using redundant time information to detect and mitigate the adverse effects of malicious nodes, which are, for example, conducting wormhole attacks [159]. In order to prevent a node to forge multiple identities by launching Sybil attacks [160], it is still necessary to have unique pair-wise keys to authenticate neighbor nodes. The security in consensus based schemes are also studied. A security filter for the average consensus method is presented in [161], and similar filters for maximum consensus approaches are studied in [162]. It should be noted that for consensus based methods, these filters act the same way as using redundant data in [158]. In summary, a secure and resilient time synchronization method requires one to utilize some redundancy in the time distribution topology to detect attacks and provide a certain level of encryption in pair-wise time data exchanges.

### 5.5. Time Synchronization in LAN and WAN

Most of the methods referenced up to this point are for PAN, and several properties of both Wireless LAN (WLAN) and WAN in Figure 1 are left out of the discussion. In this subsection, we briefly review the available methods for these networks.

In case all the nodes in an IoT deployment are connected to the Internet, in principle, it is possible to synchronize to the Internet time using NTP [11]. For mobile devices, the tailored version of NTP, referred to as SNTP [12] is mostly preferred. However, for wireless connections, varying channel conditions greatly degrade their performance as reported by Mani et al. [163]. Based on the findings, an improved version of SNTP, MNTP, that takes into account the wireless channel conditions is presented in [163]. A time synchronization scheme that is an alternative to SNTP and MNTP for high-end nodes is presented in [164]. Although these schemes can be used when Internet-based time synchronization can meet the accuracy requirements, alternative implementations discussed below are preferable.

The time synchronization for WLAN can be built upon the synchronization method known as Time Synchronization Function (TSF), defined in the IEEE 802.11 PHY specification [33], and there is a number of published works that aim at extending the possibilities enabled by this function [46]. The TSF requires all nodes to synchronize to the fastest clock in their neighborhood. It does not require rate synchronization but only offset compensation. In its bare form, TSF does not scale well so that a number of alternative schemes have been proposed in the literature. A method that estimates the clock rate is presented by Pande et al. [165], and it is shown that the clock rate estimation improves the TSF scalability.

Another widely used option for LAN is to consider time synchronization methods primarily developed for wired networks, i.e., NTP and IEEE 1588 (a.k.a. PTP). It has been shown that the implementation of NTP on WLAN yields much worse synchronization accuracy compared to wired networks [166], although the performance can be improved by moving the time-stamping closer to the hardware [167,168]. The PTP is designed for wired networks [169], and it can achieve very high accuracy. It is shown by Cooklev et al. [170] that there are certain signals in PHY layer that allow the IEEE 1588 implementations over WLAN. It is shown in the work by Kannisto et al. [171] that when timestamping is done in the hardware, nanosecond accuracy, required by the IEEE 1588 standard, can be achieved. In summary, a high accuracy synchronization can be achieved only if the time error sources can be compensated for. This line of reasoning has led some researchers to study WLAN based PTP implementations in industrial settings. The works by Lam et al. [172] and Shreshta et al. [173] aim at improving the clock precision of the IEEE 1588 to meet the stringent requirements of industrial applications. Cena et al. [174], on the other hand, take an alternative approach by proposing a reference broadcast-based scheme for IEEE 802.11 networks in infrastructure mode. This scheme can achieve high accuracy time synchronization by using software-only modifications, which is very attractive for applications implemented using commodity devices.

For WAN cellular networks, the user entities are mostly mobile, which require a time synchronous operation within the network. That is needed to maintain a certain quality of service when the user is being transferred among neighboring cells, or to keep the interference level of neighboring cells to each other under a certain value [175]. The time synchronization of the base stations is traditionally maintained using the accurate time information provided by the GPS receivers [176,177]. However, due to spatial limitations imposed by such an approach, using the IEEE 1588 for the purpose is an attractive option [177] after solving several practical problems [176]. This type of synchronization is also referred to as *backhaul* synchronization, which is different than *fronthaul* synchronization. The latter can be considered as time-sensitive networking [178] between the core network and the radio equipment. Similar synchronization issues are also common in small cell deployments [179]. Due to the similarities of these systems with time-sensitive networking over bridged-networks [180], recently the same standardization body has released a new standard on time-sensitive networking for fronthaul [181]. It is to be noted that these solutions are all based on IEEE 1588, and are optimized for the specific application. Therefore, WAN time synchronization is based on either GPS (for backhaul) or IEEE 1588.

The synchronization of WAN nodes in a cellular system is maintained by the frame alignment between a base station and network entities using the timing advance mechanism of the MAC layer [182]. This functionality is limited, and for 5G and beyond networks supporting ultra-reliable and low-latency communications (URLLC) [183] feature require tightly synchronized operation of networks entities in different cells. In particular, industrial and smart grid applications require an over-the-air time synchronization as has recently been elaborated on by Mahmood et al. [184]. The work concludes that URLLC networks can support time synchronized operation by enabling IEEE 1588 based signaling between the nodes and the base stations.

The Low-Power Wide Area (LPWA) networks enable low-power and long-range data connectivity at the expense of low data rate and higher latency [185]. They aim at providing data connectivity for IoT applications that are delay tolerant, do not need high data rates, and typically require low-power consumption (for extended operation time using finite energy source), and low cost. There are a number of 3GPP standards that are LPWA solutions (https://www.3gpp.org/news-events/1805-iot_r14), which can benefit from the time synchronization functions of the cellular networks. On the other hand, the LPWA standards operating on ISM bands, specifically, LoRaWAN (https://lora-alliance.org/resource-hub/what-lorawanr) and Sigfox (https://www.sigfox.com/en/what-sigfox/technology), have gained a lot of attention in both academia and industry. Considering the more open nature of the standard, LoRaWAN is more attractive compared to Sigfox. The limitations of LoRaWAN is investigated by Adelantado et al. [186], and it is concluded that a new channel hoping mechanisms and TDMA type channel access may overcome reliability and latency constraints, which in turn, may open up new use cases and improve flexibility. These two limitations have not gained much of interest until recent work by Ramirez et al. [187]. In the work, a CDA for two-way message exchanges is presented, and it is experimentally demonstrated that low-power nodes can maintain time synchronous operation. A time-synchronized channel hopping MAC layer implementation on top of LoRa PHY is presented by Haubro et al. [188], in which the authors have not reported a time synchronization mechanism in the effort, and relied on synchronous operation provided by the MAC. The work by Singh et al. [189] demonstrates that a coherent notion of time can be disseminated in a LoRa network, which in turn enables secure channel hopping and tighter guard times for preserving energy. The alluded works demonstrate the benefits of time synchronization in LoRa networks by showing that the technology can be used in some use cases that is not possible in bare LoRa implementation. However, the full potential of time-synchronized LoRa deployments is yet to be explored.

### 5.6. Summary

In this section, several time synchronization protocols for WSNs are summarized, and their properties are described. In particular, after defining the general error sources and timestamping methods, different messaging and multi-hop schemes are presented. Thereafter, the reference clock source, synchronization period and security related solutions are summarized. It should be noted that the methods presented in this section are comprehensive, but by no means complete. It is merely a snapshot of the available solutions that may be used by the IoT developers.

A condensed summary of the methods covered in this section is given in Table 9. It can be seen from the table that most of the synchronization methods assume a spanning-tree based multi-hop method with one-way message dissemination. Among these methods, FCSA is an important advancement to FTSP due to its improved multi-hop performance. Also, due to the widespread acceptance of IEEE 1588 as a de-facto method for wired networks, if it is possible it should be used in LAN and WAN. It is to be noted that for most of the applications, it is not required to disseminate the time coherence starting from the WAN components, as LAN networks can easily acquire a global notion of time either from the GPS or from the WAN time services.

## 6. Examples

In this section, two examples of the time data acquired by two different systems are presented.

### 6.1. A Low-Power and Short-Range Network Synchronization

In order to exemplify a practical time data acquired using IEEE 802.15.4-2006 compliant networks, we have deployed three nodes with hardware and software components described in detail in [190]. One of the nodes acts as a transmitter only, and timestamps the outgoing frames just before starting transmission. Two receivers receive that frame, and both of them store the local time using a hardware feature that enables acquiring the local time when the SFD of the frame is detected. One of the receivers transmit the received frame after pausing for 10 milliseconds. The other receiver captures also that frame in the same manner. The timer hardware has a granularity of 1/32 microseconds, and is used as a time source by all the nodes.

The time synchronization results obtained by using the data acquired by the system described above are given in Figure 21. In the figure, the time reports of the transmitter node are denoted by $C_{1}\left(t_{k}\right)$, the first receiver node by $C_{2}\left(t_{k}\right)$, and the second receiver node by $C_{3}\left(t_{k}\right)$. There are three series for each pair, and the exponentially weighted recursive estimator derived in Section 4 is used for compensating for the time variations, and the obtained time differences are visualized in Figure 21b–d. These results show that the algorithm converges in a few iterations. However, for the time data between $C_{2}\left(t_{k}\right)$ and $C_{3}\left(t_{k}\right)$, the convergence takes more iterations. This is due to the variation of the clock skew as can be observed from the figures.

The results depicted in Figure 21 show different aspects of the clock synchronization problem including the importance of hardware support for timestamping. It also validates the assumptions made in the developments in Section 4. When the relative clock skew is constant, the recursive algorithm achieves very high synchronization accuracy. However, the frequency drift is unavoidable, and it degrades the performance of the algorithm. One option that can be explored is to use recursive Bayesian filtering, e.g., Kalman filter, to estimate both clock skew and frequency drift jointly. This way, the time-relation model can be better estimated, and the estimation bias can be kept very low.

### 6.2. A Bluetooth-LE Star Network

One typical IoT deployment scenario is to interconnect low-power gadgets to the Internet using a more powerful gateway. In this example, a Bluetooth LE device is programmed to broadcast its local time report over the advertisements, while a Raspberry Pi device acquires these advertisements along with its local time. For implementation, only the embedded software of the node is modified to send the local time, where the timestamps are not acquired in the PHY layer or in the hardware. The gateway implementation is kept only in the application layer, and no kernel level or device driver related modifications are made.

The acquired data and time synchronization result are shown in Figure 22. As can be observed, using application layer timestamping induces a large number of uncertainty in the time data, and the performance of the CDA is greatly degraded. A better approach would be to use two-way message exchanges and move the timestamping closer to the hardware. However, it should also be noted that some of the IoT applications may require a loose synchronization so that the result depicted in Figure 22b is acceptable. For these cases, the time synchronization problem can be readily solved without any low-level software modifications.

## 7. Conclusions

In this article, the time synchronization problem among heterogeneous entities of Internet of Things (IoT) deployments is considered. The time synchronization is very important for IoT applications requiring chronological information ordering or synchronous execution, and also for low-power networking and transmission scheduling. It is discussed that the existing contemporary solutions such as Network Time Protocol, and available methods for wireless sensor networks can be applied only for a specific application scenario. The time synchronization problem is derived starting from the properties of the driving oscillator. The relation between time reports of two clocks is also derived, and various simplifying assumptions are presented. A number of clock discipline algorithms are introduced, and their performances are compared based on their synchronization accuracy, and computational and memory requirements. An efficient and consistent clock discipline algorithm is developed, and its performance is evaluated using both simulation and empirical data. A comprehensive survey of the time synchronization messaging protocols, and associated practical problems and their solutions are presented. The synchronization methods for local-area and wide-area networks are summarized. From the summary, it can be concluded that the precision time protocol can be used in various components of these networks, and it is expected to be an integral part of emerging communication technologies.

The most important conclusion of the presented time synchronization components is that there is no single solution that can solve all the problems, but several components can be combined to reach an optimal method for a deployment. The presented time synchronization methods are comprehensive, and there are several options for each component. Yet, there are several open research problems. The time synchronization requirements for emerging network-calibrated clocks for crystal-free radios and ultra-low power ambient back-scattering communication devices need to be defined. Although the security issues of the personal area networks of resource constraint devices are active research subjects, secure communication over loosely synchronized networks of emerging ultra-low-power entities requires more attention from the community. The presented components assume that the IoT deployments have almost independent wide-area, local-area, and personal-area networks, nevertheless, seamless integration of the time synchronization functions of these networks is an open problem. In particular, extending the precision time protocol for local area networks toward the personal area devices is an attractive option for heterogeneous IoT deployments. This and similar options enabling software reuse in different IoT devices should be further investigated, and related bottlenecks should be identified. The synchronization protocols and clock discipline algorithms for low-power and wide-area networks, such as LoRa, are in their infancy and different extensions need to be explored. The IoT researchers working on the just mentioned problems and the practitioners deploying IoT networks can use the described concepts and base their solutions on the components presented in this article.

## Figures and Tables

**Figure 1 sensors-20-05928-f001:**
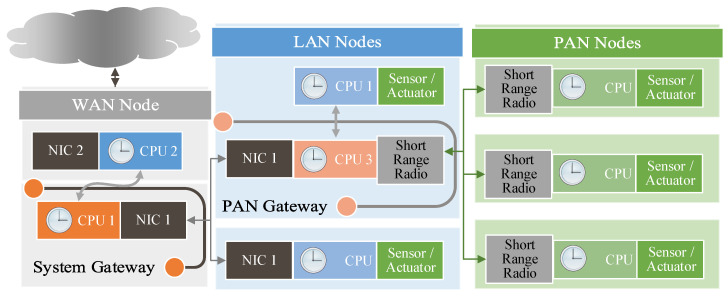
An Internet of Things deployment utilizing a (Wireless) Sensor Network to interact with the physical world. A Wide Area Network (WAN) node on the left is connected to the Internet over a Network Interface Controller (NIC), and connected to the Local Area Network (LAN) at the center using another NIC. One of the LAN nodes is connected to the Personal Area Network (PAN) on the right, which realizes the Sensor Network. Each processor in the system has its own notion of time due to different clock implementations. The network entities acting as gateways are explicitly shown.

**Figure 2 sensors-20-05928-f002:**
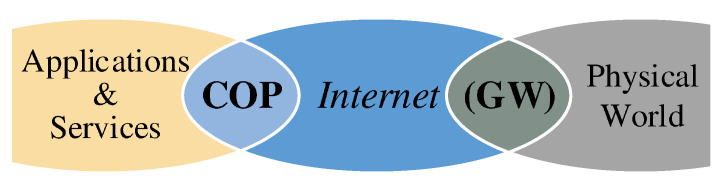
Internet of Things concept using the Internet as a link between (physical) objects, and novel application and services using a common operational picture (COP).

**Figure 3 sensors-20-05928-f003:**
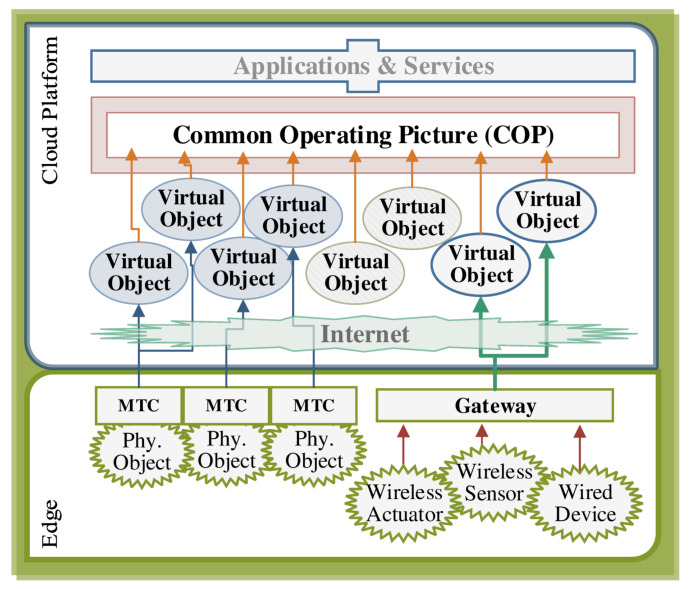
An Internet of Things platform with physical objects deployed at the edge. Some objects are connected to the Internet using machine-type communication (MTC) technologies whereas some are connected over a gateway. In either case, they are represented as virtual objects in the common operation picture, which is used by application and services.

**Figure 4 sensors-20-05928-f004:**
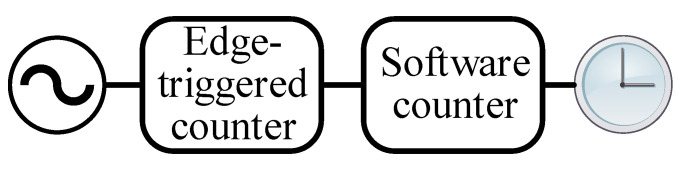
An implementation of a clock.

**Figure 5 sensors-20-05928-f005:**
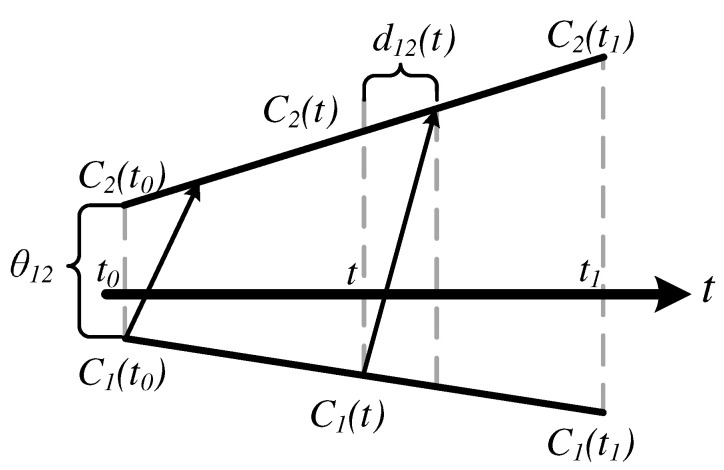
The relation between the reports of two clocks $C_{1}\left(t\right)$ and $C_{2}\left(t\right)$.

**Figure 6 sensors-20-05928-f006:**
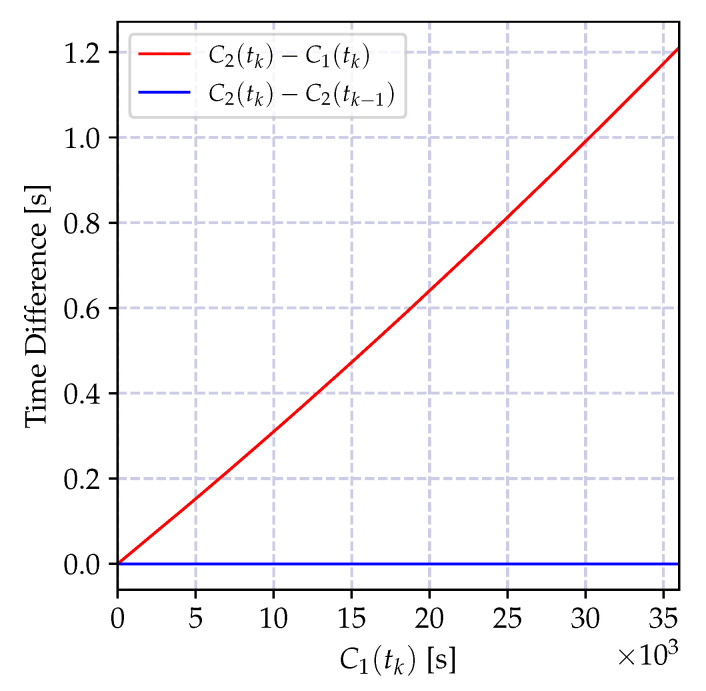
The time difference in seconds between the simulated local clock $C_{2}\left(t_{k}\right)$ and the reference time reports $C_{1}\left(t_{k}\right)$ delivered through a Gaussian message delivery delay which are associated with the universal time $t_{k}$.

**Figure 7 sensors-20-05928-f007:**
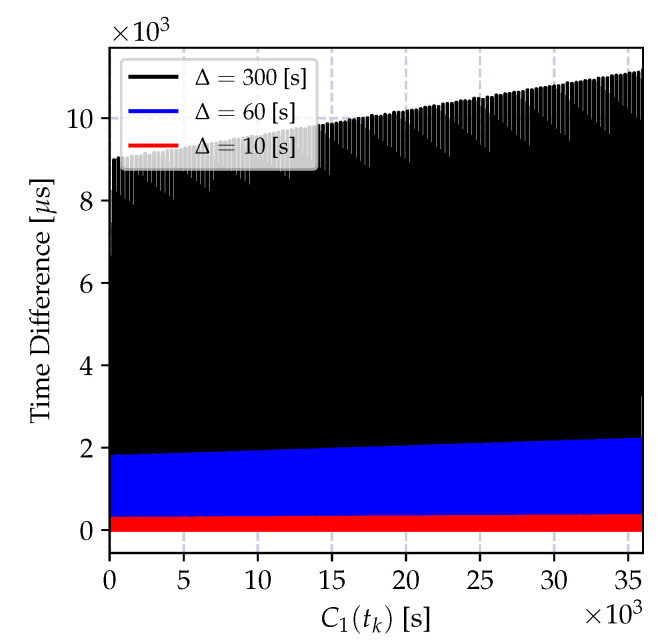
The time difference in microseconds between the time reports of a local clock and a reference clock when the time-offset is estimated at different periods Δ.

**Figure 8 sensors-20-05928-f008:**
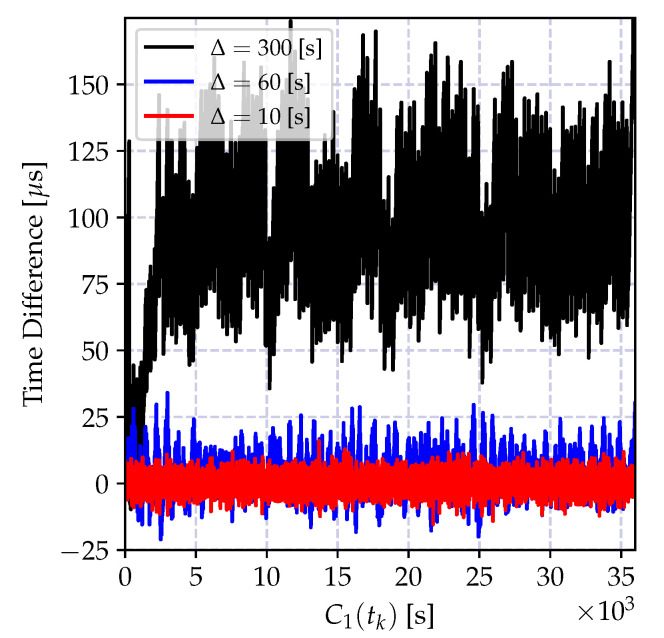
The time difference in microseconds between the time reports of a local clock and a reference clock when only the time relation parameters of the progressive time model with the message delivery delay are estimated at different periods Δ using linear regression with a regression table of N=8 entries.

**Figure 9 sensors-20-05928-f009:**
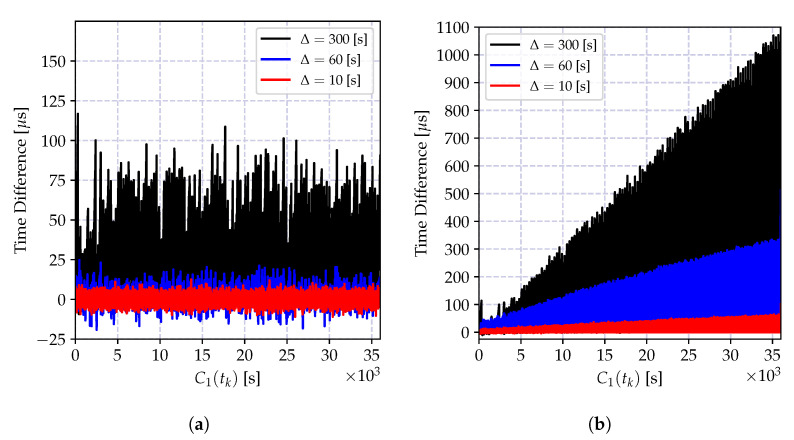
The time difference in microseconds between the local clock time reports and the reference clock reports when the time relation parameters of the incremental time model are estimated at different periods Δ using different estimators. In (**a**), the least squares estimator using a regression table of N=8 entries, and in (**b**), the recursive maximum likelihood estimator in Equation (Equation 38).

**Figure 10 sensors-20-05928-f010:**
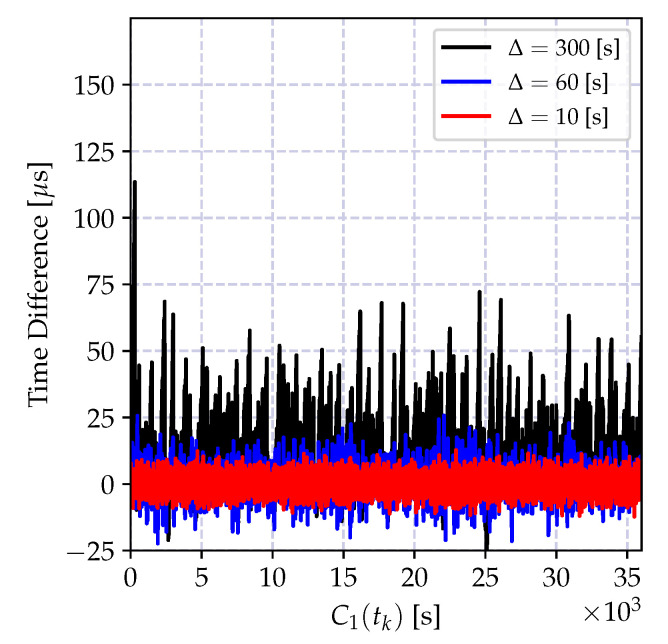
The time difference in microseconds between the time reports of a local clock and a reference clock when the time relation parameters of the progressive time model only with the message delivery delay are estimated at different periods Δ using exponentially weighted recursive estimator in Equation (39a–c) with weight λ=0.4.

**Figure 11 sensors-20-05928-f011:**
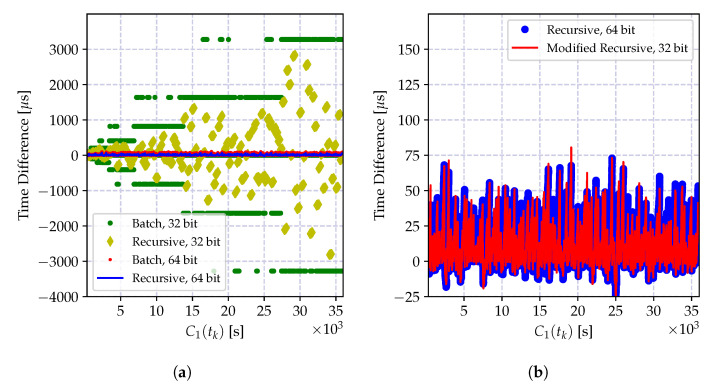
The variation of the time difference in microseconds between the time reports of a local clock and a reference clock with 32-bit and 64-bit floating point numerical precision representation of the quantities when the time relation parameters of the incremental time model are estimated every Δ=300 s. In (**a**), the performance of batch linear regression with a regression table of N=8 entries, and recursive weighted maximum likelihood estimator with weight λ=0.4. In (**b**), numerically stabilized estimator and 64-bit implementation of the recursive estimator.

**Figure 12 sensors-20-05928-f012:**
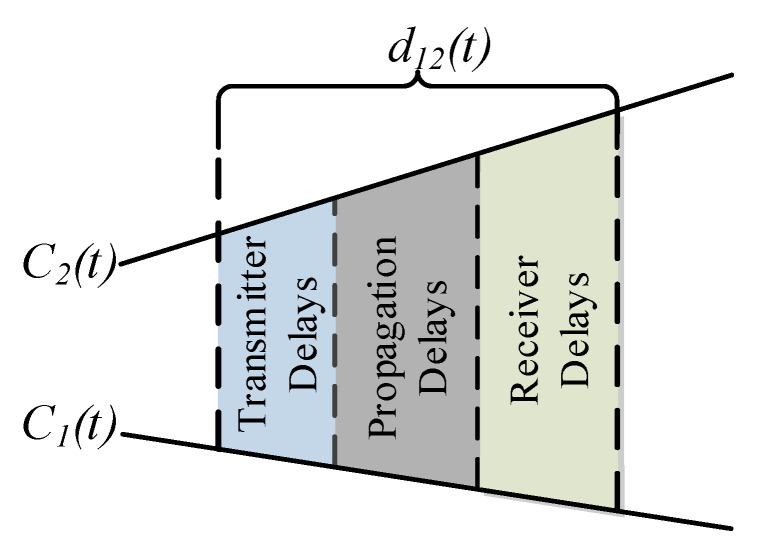
Three groups of messaging time error sources: transmitter side delays, propagation delays, and receiver side delays.

**Figure 13 sensors-20-05928-f013:**
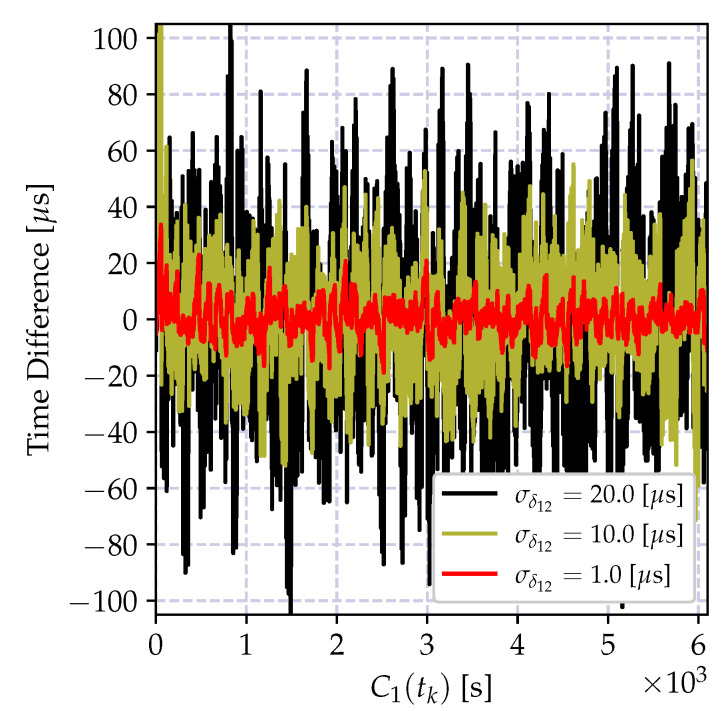
The variation of time difference in microseconds between the time reports of a local clock and a reference clock with standard deviation of zero-mean Gaussian stochastic messaging delay, denoted as $\sigma_{\delta_{12}}$. The time relation parameters of the incremental time model are estimated every Δ=10s using recursive weighted maximum likelihood estimator with weight λ=0.4.

**Figure 14 sensors-20-05928-f014:**
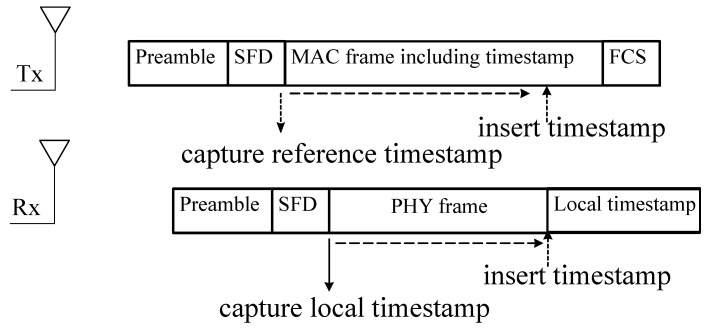
MAC layer timestamping with a IEEE 802.15.4-2006 compliant transmitter and receiver pair.

**Figure 15 sensors-20-05928-f015:**
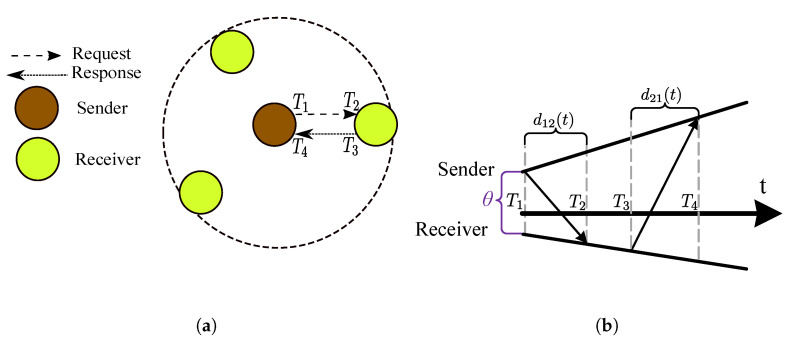
Two-way messaging scheme, where a node initiates a synchronization message exchanges by sending a synchronization request to another node while also storing its time report locally. The receiver responds back to the sender with its time report. In (**a**), a typical operation of the scheme, and in (**b**), the timeline of message exchanges.

**Figure 16 sensors-20-05928-f016:**
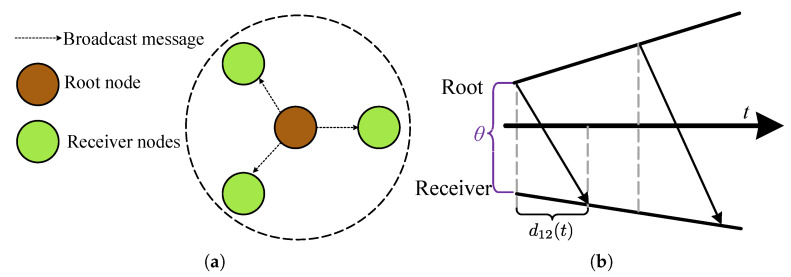
One-way messaging scheme, where a root node emits a synchronization message, and all of its neighbors receive. Each receiver synchronizes to the time of the root node. In (**a**), a typical operation of the scheme, and in (**b**), timeline of message exchanges.

**Figure 17 sensors-20-05928-f017:**
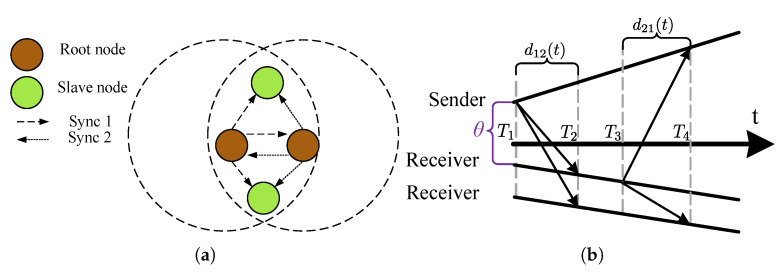
Receiver-only messaging scheme, where a pair of nodes exchanges synchronization message, and all of their common neighbors overhear this communication. Each receiver synchronizes to the time of these nodes. In (**a**), a typical operation of the scheme, and in (**b**), timeline of message exchanges.

**Figure 18 sensors-20-05928-f018:**
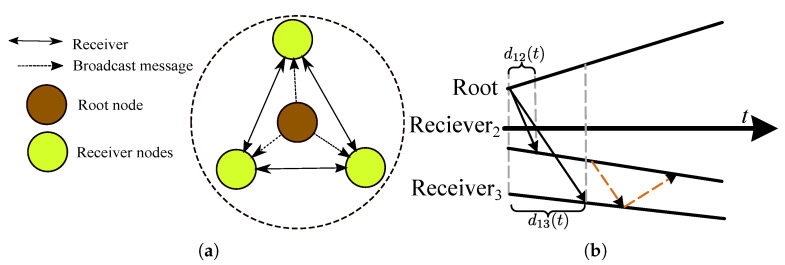
Receiver-receiver messaging scheme, where a root node emits a synchronization message, and all of its neighbors receive. Then, the receiver nodes exchange their local time along with the received reference time. In (**a**), a typical operation of the scheme, and in (**b**), timeline of message exchanges.

**Figure 19 sensors-20-05928-f019:**
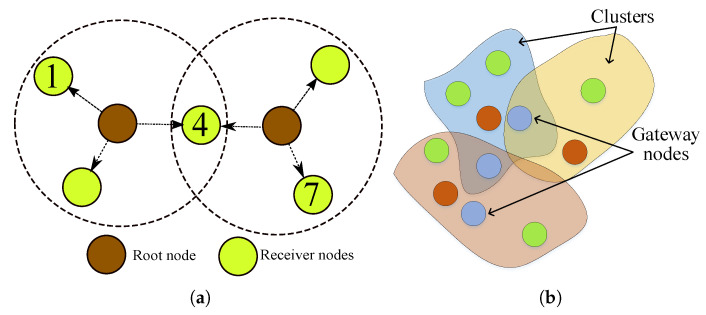
Multi-hop synchronization using receiver-receiver messaging scheme. In (**a**), two-hop messaging among nodes 1 and 7, which receive broadcasts from different root nodes. And in (**b**), a more general synchronization scenario of three clusters.

**Figure 20 sensors-20-05928-f020:**
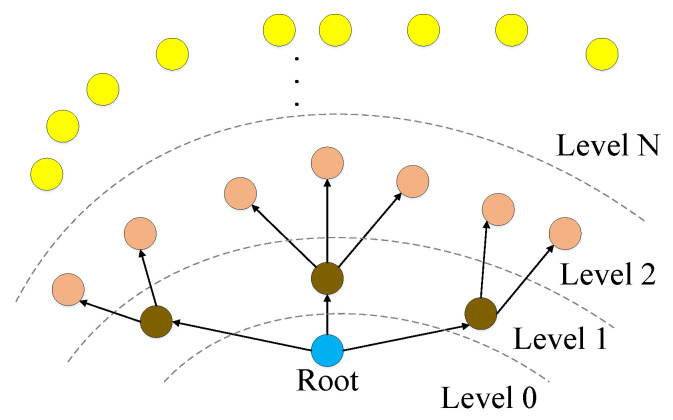
Spanning-tree based multi-hop synchronization forming a hierarchical tree of the nodes starting from the root node.

**Figure 21 sensors-20-05928-f021:**
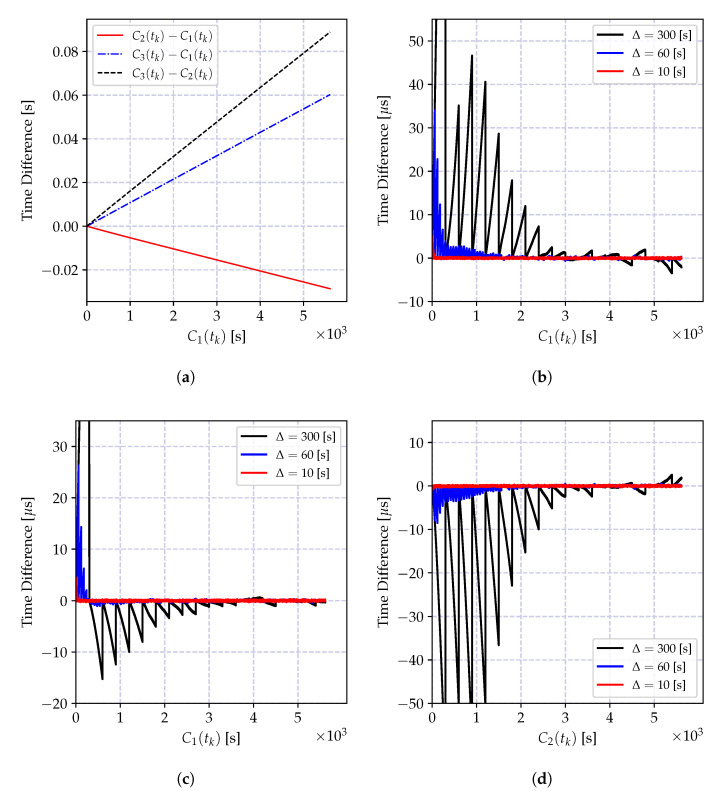
The results for the time data acquired using three IEEE 802.15.4-2006 compliant radios. In (**a**), the variation of time difference with the transmitter node’s time. In (**b**–**d**), the time difference in microseconds between the local clock time reports and the reference clock reports when the time relation parameters of the incremental time model are estimated at different periods Δ using an exponentially weighted recursive estimator in Equation (39a–c) with weight λ=0.4. The time differences between $C_{2}\left(t_{k}\right)$ and $C_{1}\left(t_{k}\right)$, $C_{3}\left(t_{k}\right)$ and $C_{1}\left(t_{k}\right)$, and $C_{3}\left(t_{k}\right)$ and $C_{2}\left(t_{k}\right)$ after compensation are shown in respective order.

**Figure 22 sensors-20-05928-f022:**
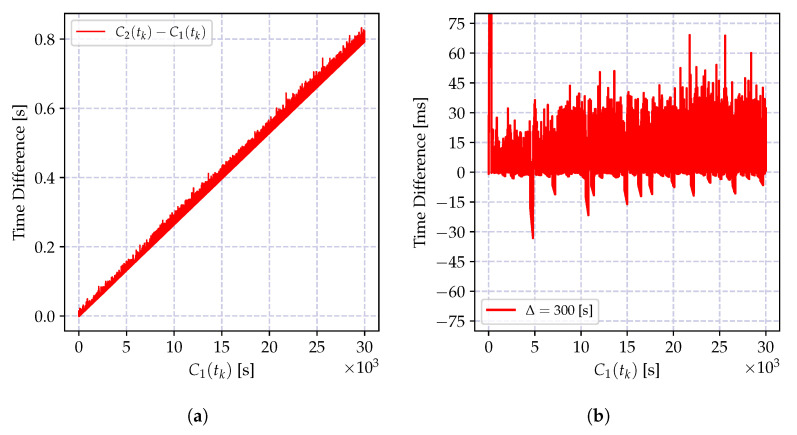
The results for the time data acquired using a Bluetooth low-energy transmitter and a Raspberry Pi gateway. In (**a**), the variation of time difference with the transmitter’s time $C_{1}\left(t_{k}\right)$. In (**b**), the time difference in milliseconds between the local clock time reports and the reference clock reports when the time relation parameters of the incremental time model are estimated every Δ=300 s using exponentially weighted recursive estimator in Equation (39a–c) with weight λ=0.1.

**Table 1 sensors-20-05928-t001:** Surveys on time synchronization in WSN.

Ref.	Year	Content
[38]	2004	An early survey on time synchronization methods in sensor networks. The work defines the problem, analyzes its requirements, and surveys available protocol till 2004.
[19]	2005	A comprehensive survey on synchronization protocols in wireless sensor networks. The survey includes synchronization methods for wired networks, and provides a detailed description of published methods till 2005. This work motivated several other articles appeared later.
[41]	2007	The earliest work that provides a set of features to classify different synchronization methods.
[43]	2008	A survey on early distributed synchronization methods for wireless networks. The work especially summarizes the coupled-clocks based network-wide synchronization approaches.
[39]	2010	A short survey of the most popular methods till 2010. The work aims at showing that by the time of writing, no synchronization method can provide security, scalability, topology independence, fast convergence and energy efficiency simultaneously.
[7]	2011	A condensed survey of WSN time synchronization in signal processing perspective. Starting from clock relation models, several clock parameter estimators are outlined. The work especially summarizes the signal processing methods for exponentially distributed delays, and related estimation methods.
[42]	2015	A classification model of time synchronization methods for WSNs. The structural, technical and global objective features of available methods are identified, and a short list of protocols are compared using the identified features.
[44]	2015	A survey of synchronization methods for machine-to-machine type communication system. A classification taxonomy for WSN synchronization is used for motivating that biologically inspired synchronicity is the most suitable option.
[40]	2019	A condensed summary of time synchronization methods for wireless sensor networks realizing an IoT deployment.

**Table 2 sensors-20-05928-t002:** Surveys on time synchronous networking.

Scope	Ref.	Year	Content
Packet switched networks	[36]	2016	A survey on standardized protocols and technologies for synchronizing devices over packet-switched networks.
Wireless LAN	[46]	2017	A survey on synchronization methods for the IEEE 802.11 (WLAN) networks in infrastructure mode.
Vehicular ad-hoc networks	[31]	2018	A survey on available methods for, and a requirement analysis of vehicular ad-hoc networks.
Cellular low latency networks	[47]	2018	A survey on technologies enabling low-latency communications in radio access networks, core network, and caching.
Ultra-low latency networks	[48]	2019	A survey on ultra-low latency networks of IEEE time-sensitive networking and IETF deterministic networking standards, along with ultra-low latency research studies of cellular networks.

**Table 3 sensors-20-05928-t003:** Time record simulation parameters.

Symbol	Value	Appearance	Description
$\theta_{1}$	1	Equation (Equation 1)	Time-offset of $C_{1}$ in seconds
$\gamma_{1}$	$10\times 10^{-6}$	Equation (Equation 1)	Frequency offset of $C_{1}$
$\omega_{1}$	$1\times 10^{-12}$	Equation (Equation 1)	Frequency drift of $C_{1}$
$c_{1}$	$1\times 10^{-8}$	Equation (2a,b)	Oscillator constant of $C_{1}$
$\theta_{2}$	2	Equation (Equation 1)	Time offset of $C_{2}$ in seconds
$\gamma_{2}$	$-20\times 10^{-6}$	Equation (Equation 1)	Frequency offset of $C_{2}$
$\omega_{2}$	$-1\times 10^{-10}$	Equation (Equation 1)	Frequency drift of $C_{2}$
$c_{2}$	$1\times 10^{-10}$	Equation (2a,b)	Oscillator constant of $C_{2}$
$D_{12}$	$1\times 10^{-3}$	Equation (Equation 5)	Deterministic and constant message delivery delay in seconds
$E\{\delta_{12}^{2}\}$	$1\times 10^{-10}$	Equation (Equation 5)	Variance of stochastic message delivery delay in seconds square

**Table 4 sensors-20-05928-t004:** Summary of clock relation models.

Model	References	Advantages	Disadvantages
**Offset-only model**	[52,53,54]	A single parameter model taking into account only the clock offset term. This is the simplest model.	It has a large modeling error bias, and cannot be used for high accuracy and low-power time synchronization purposes.
**Progressive linear model only with delivery delay**	[7,55,56,57,58,59,60]	A first order time relation model that can be used for maintaining energy efficient time synchronous operation. This model is the most widely accepted model in the literature.	It does not take into account the time variation of the clock-skew parameter and oscillator-induced time correlations, which upper bounds the synchronization period so that frequent time report exchange is required.
**Incremental linear model only with delivery delay**	[61,62]	A linear model of clock skew, which enables a dynamical model for clock skew.	It does not depend on clock offset, and does not take into account the oscillator-induced correlations. A two step clock discipline algorithm is required.
**Higher order progressive models only with delivery delay**	[61,62,64,65]	A higher order (with respect to time argument) model which takes into account the dynamics of the clock skew. Enables low-power time synchronization by prolonging synchronization periods.	The number of parameters are increased, which increases the required amount of computational resources. For the models with degree higher than two, physical clock terminology cannot be used.
**Incremental linear model with delivery delay and oscillator-induced correlation**	[50]	An oscillator-induced time correlation compensated model, which enables high accuracy time synchronization.	It does not depend on clock offset. A two step clock discipline algorithm is required.

**Table 5 sensors-20-05928-t005:** Statistics of CDA results.

CDA	Δ Seconds	Mean Microseconds	Standard Deviation Microseconds	Skewness
**Offset-only Figure 7**	10	151.32	97.28	0.037
	60	991.99	586.87	0.039
	300	5025.82	2935.20	0.040
**Batch least squares Figure 8**	10	0.097	3.06	−0.015
	60	3.92	7.18	0.120
	300	94.85	29.52	−0.079
**Batch least squares Figure 9a**	10	0.02	2.37	0.056
	60	1.36	5.09	0.254
	300	33.58	23.73	0.402
**Recursive MLE Figure 9b**	10	15.59	13.72	0.882
	60	93.58	75.99	0.884
	300	271.79	241.65	0.979
**Weighted recursive MLE Figure 10**	10	0.001	2.62	0.062
	60	0.53	5.50	0.155
	300	13.40	15.35	1.073

**Table 6 sensors-20-05928-t006:** Available clock discipline algorithms

Estimator	Model	Advantages	Disadvantages
**Offset-only estimation** [52,53,54]	(Equation 17)	A single parameter estimator, which assumes $\alpha_{12}=1$. It is used for adjusting the time using two-way message exchanges.	It has a large modeling error bias, and cannot be used for high accuracy synchronization purposes for low-power networks.
**Batch least squares estimator** [7,55,56,57,58,59,60]	(Equation 18)	A simple table-based linear regression estimator. It estimates both skew and offset parameter, and has an acceptable performance. High-precision numerical values are needed to achieve the reported performance.	Its model does not take into account the time variation of clock-skew parameter and oscillator-induced time correlations, which upper bounds the synchronization period. It requires re-calculation of the estimates using all the time values in the table whenever a new time report is available. It treats the clock-skew as an unknown constant.
**Adaptive skew estimator (Bayesian)** [61,62,63]	(Equation 19)	A linear state estimator for clock skew, which takes into account the frequency drift. Multiple skew dynamical models can be used simultaneously to account for different practical situations. It assumes dynamics of clock-skew given in Equations (Equation 31), (Equation 32) or (Equation 33)	It does not depend on clock offset, and does not take into account the oscillator-induced correlations into account. Several computational steps of Kalman Filter are required to update its skew estimate. The underlying model requires non-obvious modifications to reach numerically stable estimates.
**Adaptive skew estimator (Closed-loop)** [71,72,73,74]	(Equation 19)	Implicit clock skew estimate using PI controller, which resembles PLL loop. Well-known control theoretical tools can be used for adjusting its gains. In its bare form it is equivalent to constant gain Kalman Filter. Adaptive version can be used to adjust its gains on-the-fly. Although not reported, it is expected to be numerically stable.	It does not depend on clock offset, and does not take into account the oscillator-induced correlations into account. A two-step clock discipline algorithm is required.
**Recursive clock skew estimator** [50]	(Equation 22)	A computationally efficient (recursive) MLE of the skew. It takes into account oscillator-induced correlations.	It does not depend on clock offset so that two-step clock discipline algorithm is required. It cannot follow the changes in the clock-skew due to frequency drift since it uses all the past time values.
**Weighted recursive clock skew estimator** [This work]	(Equation 22)	A computationally efficient (recursive) MLE of the skew. It takes into account oscillator-induced correlations. It supports numerically stable implementation given in Equation (Equation 41). It can follow the dynamical variations in the clock skew by limiting number of measurements affecting the estimates.	It does not depend on clock offset so that two step clock discipline algorithm is required. It requires adjusting the forgetting factor λ for each deployment. This parameter can be selected to include 2–3 reports, since it converges to the correct skew very fast. Numerically stable version requires a constant gain parameter *K*. This gain can be selected to move significant bits of clock-skew. In practice, $K=1\cdot 10^{6}$ can be used.

**Table 7 sensors-20-05928-t007:** Comparison of clock discipline algorithms.

Name	Model	Type	Skew Estimate	Offset Estimate	Bias	Complexity	Sampling
**Offset-only**	Progressive	Batch	${\hat{\alpha }}_{12}=1$	(Equation 27)	(Equation 28)	O(1)	
**Batch least squares**	Progressive	Batch	([Disp-formula FD29a-sensors-20-05928])	(29b)	(Equation 30)	O(N)	Periodic
**Batch least squares**	Incremental	Batch	([Disp-formula FD35a-sensors-20-05928])	(35b)	(Equation 30)	O(N)	Periodic
**Recursive maximum likelihood**	Incremental	Recursive	(Equation 38)	(35b)	(Equation 30)	O(1)	A-periodic
**Weighted recursive maximum likelihood**	Incremental	Recursive	(39a–c)	(35b)	(Equation 30)	O(1)	A-periodic
**Numerically-stable recursive maximum likelihood**	Incremental	Recursive	(Equation 41)	(35b)	(Equation 30)	O(1)	A-periodic

**Table 8 sensors-20-05928-t008:** Messaging error sources.

**Transmitter Delays**	*Message Processing*	*Frame Preparation*	*Software Delay*	*Encoding Time*	*Calibration Time*	*access Time*	*Transmission Time*
	deterministic	deterministic	random	deterministic	random	random	deterministic
**Propagation** **Delays**	*propagation time*						
	deterministic						
**Receiver** **Delays**	*reception time*		*decoding time*	*byte alignment time*		*interrupt handling time*	
	deterministic		deterministic	deterministic		random	

**Table 9 sensors-20-05928-t009:** A summary of time synchronization protocols.

Synchronization Protocol	Ref.	Messaging Scheme	Multi-Hop Scheme	Description
Network Time Protocol (NTP)	[10,11]	two-way	spanning-tree	NTP is the de-fact time synchronization protocol for large networks such as the Internet. The achievable performance is lower than demands of certain applications.
IEEE 1588: Precision Time Protocol (PTP)	[13]	two-way	spanning-tree	A high precision time synchronization standard for measurement equipments. Nowadays, it is used in all wired networks applications demanding high precision time synchronization.
Time-synchronization Protocol for Sensor Network (TPSN)	[82]	two-way	spanning-tree	TPSN is one of the early multi-hop time synchronization protocols for WSN. It doesn’t compensate for the clock skew.
Reference Broadcast Protocol (RBS)	[55]	receiver-receiver	cluster-based	RBS achieves high synchronization accuracy by eliminating transmitter side errors. It enables implementations with minimal modifications in the lower-level software components.
Flooding Time Synchronization Protocol (FTSP)	[56]	one-way	spanning-tree	FTSP is widely accepted as the de-facto synchronization protocol for WSN. It achieves high synchronization accuracy using MAC layer timestamping.
Flooding with Clock Speed Agreement (FCSA)	[59]	one-way	spanning-tree	FCSA is a slow-flooding based multi-hop solution that extends FTSP. It forces the network to reach clock-speed agreement by combining flooding with a distributed component.
Robust Synchronization (R-Sync)	[37]	receiver-only	spanning-tree	R-Sync provides low-power and robust time synchronization protocol for industrial applications.
Time Diffusion Protocol (TDP)	[106]	two-way	synchronous diffusion	TDP aims at providing a protocol resilient to node failures and topology changes while taking into account the energy constraints. It is the predecessor of distributed algorithms.
Distributed Time Synchronization Protocol (DTSP)	[108]	one-way	distributed	DTSP is the earliest distributed time synchronization method. It uses spatial averaging to converge to a common notion of time.
Gradient Time Synchronization Protocol (GTSP)	[109]	one-way	distributed	GTSP is a time synchronization protocol making use of the gradient property. It performs local spatial averaging to reach to a common notion of time.
Average Time Synchronization (ATS)	[114]	one-way	distributed	ATS is the first consensus-based time synchronization method for WSN. It tries to reach to the time consensus by converging to a common offset and clock-rate values.
Reachback Firefly Algorithm (RFA)	[128]	one-way	distributed	RFA provides a mechanism for network-wide synchronicity, which does not provide a coherent notion of time, but only an option for synchronously executing of a certain task. It is inspired by synchronicity in biological systems.

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
