# Peer review of "Overview of Time Synchronization for IoT Deployments: Clock Discipline Algorithms and Protocols"

_sensors, 2020, doi:10.3390/s20205928_

Round 1
Reviewer 1 Report
This is a voluminous tutorial-type piece of work with a very comprehensive list of 190 references. The authors have treated the topic in a very cohesive way, highlighted a number of fundamental challenges, reviewed many approaches in a well-structured way, used a common notation, took up a series of practice-relevant issues, and provided a set of relevant examples.
Line 446: which is the basis for the recommended value of K?
Line 481: "clearly implies" -- not really. The table does not communicate the superiority without extra explanations.
The reviewer proposes a set of minor modifications:
- Refer to "article" instead of "paper" (e.g. lines 14, 1224)
- "Exponentially" (line 159); "Exponential" (line 251) => "exponentially"
- There are several issues with missing plural-s (e.g. line 167: present => presents)
- Wording, line 168: "from bottom to up" => "in a bottom-up"
- Missing articles, e.g. line 218: "time" => "the time"; line 289: "offset-only" => "the offset-only"; line 375 below (32) => "or the constant"; line 446 => "the scaled clock..."; lines 1006+1007: => "a GPS"; line 1097: => "an alternative"
- Missing hyphen, e.g. line 281: "well known" => "well-known"; line 370 => "oscillator-induced"; line 700: => "standard-compliant; line 701/702: => "receiver-only"; line 965+969: => "trade-off"; line 1047: => "denial-of-service"; line 1054/1055: => "well-known"; line 1122: "software-only"; line 1124: => "time-synchronous"
- Missing blank, line 292 ahead of "Although"
- Wording, line 375 just before (32) => "the discretized version of which"
- Singular/plural issue, line 411: "are" => "is"; line 476: "it is" => "they are "; line 997: "yield" => "yields"; line 1043: "type" => "types"; line 1047: "solution" => "solutions"; line 1102: => "aim"
- Figure 13: The curves for 10.0 and 5.0 microseconds are extremely difficult to discern
- Lines 595-596: M/M/1 is a standard approach, but reality is probably much more complicated.
- Wording, line 765: "can be in" => "is expected to be"
- Line 1010: "at al." => "et al."
- Line 1014: "mains" => "main" (?)
- Lines 1052-1053: Rephrasing needed
- Lines 1080: "forge multiple of identity" -- something missing?
- Line 1173: "no means" => "by no means"
- Line 1200: fragment of a sentence at the end of the line
- Figure 22: The curves for 300 and 60 s are hardly visible in the figure
By the way, to which extent can one rely upon correct readings, and what would be the consequences for the estimation? Hardware-based time stamping as mentioned a.o. in line 1115 is usually rather reliable, while software-based stamping can introduce huge errors (see a.o. the work by Arlos on quality of measurements in Internet). This goes probably beyond the scope of this work
Author Response
Dear reviewer,
We would like to express our sincere gratitude for contributing to the volunteer review process of our article. We believe your suggestions have improved the presentation quality of the article. Below, you can find the responses to your comments.
- Line 446: which is the basis for the recommended value of K?
Answer: The text is modified to clarify how K is selected.
The value of $K$ is selected based on the expected variation of clock skew between synchronization periods. If the environment is stable, the clock skew $\alpha$ stays in a close neighborhood of $1$, and $K$ is selected large if the CDA is required to react to small changes. In a
more unstable environment, a smaller $K$ value may be used. For example, $K=1\cdot10^6$ can be used to cover most of the cases encountered in practice. - Line 481: "clearly implies" -- not really. The table does not communicate the superiority without extra explanations.
Answer: The sentence containing “clearly implies” now extended to explain the reason why the method is superior to the others as follows:
The recursive estimators are appealing for their lower computational and memory requirements. An algorithm that is a-periodic can handle occasional packet drops, which is common in wireless networks. The results given in Table 5 and Table 7, and the comparison of the methods in Table 6 imply that the numerically stable estimator achieves a higher accuracy time synchronization while using a lower amount of computational resources and providing solutions for numerical problems. Therefore, it is superior to the other estimators presented in this section. - Grammatical errors:
- Refer to "article" instead of "paper" (e.g. lines 14, 1224)
- "Exponentially" (line 159); "Exponential" (line 251) => "exponentially"
- There are several issues with missing plural-s (e.g. line 167: present => presents)
- Wording, line 168: "from bottom to up" => "in a bottom-up"
- Missing articles, e.g. line 218: "time" => "the time"; line 289: "offset-only" => "the offset-only"; line 375 below (32) => "or the constant"; line 446 => "the scaled clock..."; lines 1006+1007: => "a GPS"; line 1097: => "an alternative"
- Missing hyphen, e.g. line 281: "well known" => "well-known"; line 370 => "oscillator-induced"; line 700: => "standard-compliant; line 701/702: => "receiver-only"; line 965+969: => "trade-off"; line 1047: => "denial-of-service"; line 1054/1055: => "well-known"; line 1122: "software-only"; line 1124: => "time-synchronous"
- Missing blank, line 292 ahead of "Although"
- Wording, line 375 just before (32) => "the discretized version of which"
- Singular/plural issue, line 411: "are" => "is"; line 476: "it is" => "they are "; line 997: "yield" => "yields"; line 1043: "type" => "types"; line 1047: "solution" => "solutions"; line 1102: => "aim"
- Wording, line 765: "can be in" => "is expected to be"
- Line 1010: "at al." => "et al."
- Line 1014: "mains" => "main" (?)
- Lines 1052-1053: Rephrasing needed
- Lines 1080: "forge multiple of identity" -- something missing?
- Line 1173: "no means" => "by no means"
- Line 1200: fragment of a sentence at the end of the line
Answer: These and other identified wording and grammatical errors are corrected. Please see the accompanying difference file for details.
- Figure 13: The curves for 10.0 and 5.0 microseconds are extremely difficult to discern
Answer: The curves now are evaluated at 1, 10, and 20 microseconds, and the color for 10 microseconds plot is modified to improve visual difference. - Lines 595-596: M/M/1 is a standard approach, but reality is probably much more complicated.
Answer: A footnote is added to clarify that in practice the delivery delay is a more complex queue than M/M/1.
The M/M/1 queue assumes a single server with exponentially distributed service times and Poisson distributed arrivals. In practice, the software delay, interrupt handling time, and access time are all independent queues, and their joint impact may require more complex statistical models. - Figure 22: The curves for 300 and 60 s are hardly visible in the figure
Answer: The lines for 10 and 60 seconds are removed since the most important evaluation is for 300 seconds. - By the way, to which extent can one rely upon correct readings, and what would be the consequences for the estimation? Hardware-based time stamping as mentioned a.o. in line 1115 is usually rather reliable, while software-based stamping can introduce huge errors (see a.o. the work by Arlos on quality of measurements in Internet). This goes probably beyond the scope of this work
Answer: In a practical implementation over the Internet, the time synchronization packets are filtered before their data are input to the estimator (CDA). One simple option is to check whether the received time data falls into a range around the expected value. Such filters and their impact are deliberately left out of the scope since they are implementation and deployment dependent.
Reviewer 2 Report
In this paper, a comprehensive review of time synchronization methods for internet of things (IoT) deployments was presented. Detailed derivations for the clock model, and various clock relation models are provided. The clock discipline algorithms are presented in a tutorial format. It is a meaningful work for study of the time synchronization problems of IoT deployments.
In general, this paper is well prepared and presented. The following minor comments may help authors to improve the paper:
- This paper mainly is a survey paper. Authors may consider to reflect it in the title;
- Possible future research directions should be discussed in the Conclusions or before the Conclusions.
Author Response
Dear reviewer,
We would like to express our sincere gratitude for contributing to the review process of our article. Below, you can find our response to your comments.
- This paper mainly is a survey paper. Authors may consider to reflect it in the title.
Answer: We modified the title of the article as follows:
Overview of Time Synchronization for IoT Deployments: Clock Discipline Algorithms and Protocols - Possible future research directions should be discussed in the Conclusions or before the Conclusions.
Answer: We significantly modified the conclusion section, and included future research directions as follows:
In this article, the time synchronization problem among heterogeneous entities of Internet of Things (IoT) deployments is considered. The time synchronization is very important for IoT applications requiring chronological information ordering or synchronous execution, and also for low-power networking and transmission scheduling. It is discussed that the existing contemporary solutions such as Network Time Protocol, and available methods for wireless sensor networks can be applied only for a specific application scenario. The time synchronization problem is derived starting from the properties of the driving oscillator. The relation between time reports of two clocks is also derived, and various simplifying assumptions are presented. A number of clock discipline algorithms are introduced, and their performances are compared based on their synchronization accuracy, and computational and memory requirements. An efficient and consistent clock discipline algorithm is developed, and its performance is evaluated using both simulation and empirical data. A comprehensive survey of the time synchronization messaging protocols, and associated practical problems and their solutions are presented. The synchronization methods for local-area and wide-area networks are summarized. From the summary, it can be concluded that the precision time protocol can be used in various components of these networks, and it is expected to be an integral part of emerging communication technologies.
The most important conclusion of the presented time synchronization components is that there is no single solution that can solve all the problems, but several components can be combined to reach an optimal method for a deployment. The presented time synchronization methods are comprehensive, and there are several options for each component. Yet, there are several open research problems. The time synchronization requirements for emerging network-calibrated clocks for crystal-free radios and ultra-low power ambient back-scattering communication devices need to be defined. Although the security issues of the personal area networks of resource constraint devices are active research subjects, secure communication over loosely synchronized networks of emerging ultra-low-power entities requires more attention from the community. The presented components assume that the IoT deployments have almost independent wide-area, local-area, and personal-area networks, nevertheless, seamless integration of the time synchronization functions of these networks is an open problem. In particular, extending the precision time protocol for local area networks toward the personal area devices is an attractive option for heterogeneous IoT deployments. This and similar options enabling software reuse in different IoT devices should be further investigated, and related bottlenecks should be identified. The synchronization protocols and clock discipline algorithms for low-power and wide-area networks, such as LoRa, are in their infancy and different extensions need to be explored. The IoT researchers working on the just mentioned problems and the practitioners deploying IoT networks can use the described concepts and base their solutions on the components presented in this article.